# Safety-Biased Policy Optimisation: Towards Hard-Constrained Reinforcement Learning via Trust Regions

## Abstract

Reinforcement learning (RL) in safety-critical domains requires agents to maximise rewards while strictly adhering to safety constraints. Existing approaches, such as Lagrangian and projection-based methods, often either fail to ensure near-zero safety violations or sacrifice reward performance in the face of hard constraints. We propose *Safety-Biased Trust Region Policy Optimisation (SB-TRPO)*, a new trust-region algorithm for hard-constrained RL. SB-TRPO adaptively biases policy updates towards constraint satisfaction while still seeking reward improvement. Concretely, it performs trust-region updates using a convex combination of the natural policy gradients of cost and reward, ensuring a fixed fraction of optimal cost reduction at each step. We provide a theoretical guarantee of local progress towards safety, with reward improvement when gradients are suitably aligned. Experiments on standard and challenging *Safety Gymnasium* tasks show that SB-TRPO consistently achieves the best balance of safety and meaningful task completion compared to state-of-the-art methods.

## 1 Introduction

Reinforcement learning (RL) has achieved remarkable success in domains ranging from games to robotics, largely by optimising long-term cumulative rewards through trial-and-error interaction with an environment. In many real-world applications, however, unconstrained reward maximisation is insufficient: agents must also satisfy safety or operational constraints. This problem is naturally formulated as a Constrained Markov Decision Process (CMDP), which is equipped with both reward and cost signals. Safe RL has been studied under a variety of objectives (see (Wachi et al., 2024; Gu et al., 2024b) for surveys). Most work formulates the problem as maximising discounted expected reward subject to the discounted expected cost remaining below a threshold (e.g., (Ray et al., 2019; Achiam et al., 2017; Yang et al., 2022)). This formulation is appropriate when costs represent resources or other soft limitations—for example, an autonomous drone may need to complete its mission before its battery is depleted. However, in many safety-critical domains any nonzero cost corresponds to catastrophic failure. Autonomous robots handling hazardous materials cannot afford any collisions, and autonomous vehicles must never exceed speed or distance limits in safety-critical zones. In such settings, a hard safety constraint is more appropriate: policies must avoid any violations entirely, rather than merely achieving low discounted cost in expectation.

**Limitations of existing approaches.** Despite significant progress, existing constrained RL methods face serious limitations in safety-critical domains. Lagrangian methods, such as TRPO- and PPO-Lagrangian (Ray et al., 2019), typically either fail to achieve meaningful task completion or high safety, but rarely succeed at both simultaneously. Projection-based methods such as Constrained Policy Optimization (CPO) (Achiam et al., 2017) explicitly attempt to maintain 0-cost policies at each update, which can severely degrade reward performance in challenging environments, as feasible updates may be rare or require very small steps. More recent algorithms, such as CRPO (Xu et al., 2021) and PCRPO Gu et al.

(2024a), switch between cost and reward optimisation, with PCRPO additionally trying to reconcile cost and reward gradients via projection and simple averaging. However, when safety is hard to achieve, these methods focus almost exclusively on safety, resulting in low task achievement, as for CPO. Overall, existing methods either violate hard safety constraints or sacrifice task performance significantly, highlighting the need for approaches that robustly achieve both high safety and high reward.

**Our approach.** We propose *Safety-Biased Trust Region Policy Optimisation* (SB-TRPO), a new algorithm for hard-constrained RL. SB-TRPO modifies the TRPO update so that each step guarantees progress towards satisfying the safety constraint while also improving reward whenever possible. Unlike CPO and related methods, SB-TRPO does *not* switch into a separate recovery phase, avoiding over-conservatism and yielding smoother learning. It also generalises CPO (recovered when enforcing maximal safety progress). Conceptually, SB-TRPO performs a dynamic convex combination of the cost and reward natural gradients, ensuring a fixed fraction of optimal cost reduction while allocating remaining update capacity to reward. This provides reliable progress on safety without unnecessarily sacrificing task performance.

**Contributions.** The main contributions of this work are as follows:

- We introduce *SB-TRPO*, a novel algorithm for enforcing hard safety constraints by dynamically biasing policy updates towards cost reduction within a principled trust-region framework.
- We provide theoretical justification for the proposed policy updates, showing that they guarantee local reduction of constraint violations without sacrificing reward improvement opportunities when gradients are well-aligned.
- We benchmark SB-TRPO against state-of-the-art safe RL algorithms on both standard and challenging *Safety Gymnasium* environments, demonstrating that competitive rewards can be achieved with low cost and high safety.

## 2 RELATED WORK

**Primal–Dual Methods.** Primal–dual methods for constrained RL solve a minimax problem, maximising the penalised reward with respect to the policy while minimising it with respect to the Lagrange multiplier to enforce cost constraints. RCPO (Tessler et al., 2018) is an early two-timescale scheme using vanilla policy gradient and slow multiplier updates. Under additional assumptions (e.g., that all local optima are feasible), it converges to a locally optimal feasible policy. Likewise, TRPO-Lagrangian and PPO-Lagrangian (Ray et al., 2019) use trust-region or clipped-surrogate updates for the primal policy parameters. PID Lagrangian methods (Stooke et al., 2020) augment the standard Lagrangian update with additional terms, reducing oscillations when overshooting the safety target is possible. Similarly, APPO (Dai et al., 2023), which is based on PPO, augments the Lagrangian of the constrained problem with a quadratic deviation term to dampen cost oscillations.

**Trust-Region and Projection-Based Methods.** Constrained Policy Optimization (CPO) (Achiam et al., 2017) enforces a local approximation of the CMDP constraints within a trust region and performs pure cost-gradient steps when constraints are violated, aiming to always keep policy updates within the feasible region. FOCOPS (Zhang et al., 2020) is a first-order method solving almost the same abstract problem for policy updates in the nonparameterised policy space and then projects it back to the parameterised policy. P3O (Zhang et al., 2022) is a PPO-based constrained RL method motivated by CPO. It uses a Lagrange multiplier, increasing linearly to a fixed upper bound, and clipped surrogate updates to balance reward improvement with constraint satisfaction. Constrained Update Projection (CUP) (Yang et al., 2022), in contrast to CPO, formulates surrogate reward and cost objectives using generalised performance bounds and Generalised Advantage Estimators (GAE), and projects each policy gradient update into the feasible set defined by these surrogates, allowing updates to jointly respect reward and cost approximations. Recently, Milosevic et al. (2024) proposed C-TRPO, which reshapes the geometry of the policy space

by adding a barrier term to the KL divergence so that trust regions contain only safe policies (see Section D.1 for an extended discussion). However, similar to CPO, C-TRPO switches to pure cost-gradient updates when the policy becomes infeasible, entering a dedicated recovery phase. In follow-up work, Milosevic et al. (2025) introduced C3PO, which resembles P3O and relaxes hard constraints using a clipped penalty on constraint violations.

**Reward–Cost Switching.** CRPO (Xu et al., 2021) is a constrained RL method that alternates between reward maximistion and cost minimistion depending on whether the current cost estimate indicates the policy is infeasible. Motivated by CRPO's tendency to oscillate between purely reward- and cost-focused updates, Gu et al. (2024a) propose PCRPO, which mitigates conflicts between reward and safety gradients using a softer switching mechanism. When both objectives are optimised, the gradients are averaged, and if their angle exceeds 90°, each is projected onto the normal plane of the other.

**Model-Based and Shielding Approaches.** Shielding methods (Alshiekh et al., 2018; Belardinelli et al., 2025; Jansen et al., 2020) intervene when a policy proposes unsafe actions, e.g. via lookahead or model predictive control, but typically require accurate dynamic models and can be costly. Yu et al. (2022) train an auxiliary policy to edit unsafe actions, balancing reward against the extent of editing. Other typically model-based methods leverage control theory (Perkins & Barto, 2002; Berkenkamp et al., 2017; Chow et al., 2018; Wang et al., 2023).

## 3 Problem Formulation

We consider a *constrained Markov Decision Process* (CMDP) defined by the tuple $(\mathcal{S}, \mathcal{A}, P, r, c, \gamma)$, where $\mathcal{S}$ and $\mathcal{A}$ are the (potentially continuous) state and action spaces, $P(s' \mid s, a)$ is the transition kernel, $r : \mathcal{S} \to \mathbb{R}$ is the reward function, $c : \mathcal{S} \to \mathbb{R}_{\geq 0}$ is a cost function representing unsafe events, and $\gamma \in [0, 1)$ is the discount factor. A stochastic policy $\pi(a \mid s)$ induces a distribution over trajectories $\tau = (s_0, a_0, s_1, a_1, \dots)$ according to $P$ and $\pi$.

In safety-critical domains, the objective is to maximise discounted reward while ensuring that unsafe states are never visited, i.e. $c(s_t) = 0$ for all times $t$ almost surely:

$$\max_{\pi} \quad J_r(\pi) := \mathbb{E}_{\tau \sim \pi} \left[ \sum_{t=0}^{\infty} \gamma^t \cdot r(s_t) \right] \quad \text{s.t.} \quad \mathbb{P}_{\tau \sim \pi} \left[ \exists t \in \mathbb{N}. \, c(s_t) > 0 \right] = 0 \qquad \textbf{(Problem 1)}$$

Directly enforcing this zero-probability constraint over an *infinite horizon* is intractable in general, especially under stochastic dynamics or continuous state/action spaces. Even a single unsafe step violates the constraint, and computing the probability of an unsafe event across the infinite horizon is typically infeasible.

To make the problem tractable, we re-formulate it as an *expected discounted cost constraint*:

$$\max_{\pi} J_r(\pi) := \mathbb{E}_{\tau \sim \pi} \left[ \sum_{t=0}^{\infty} \gamma^t \cdot r(s_t) \right] \quad \text{s.t.} \quad J_c(\pi) := \mathbb{E}_{\tau \sim \pi} \left[ \sum_{t=0}^{\infty} \gamma^t \cdot c(s_t) \right] = 0. \qquad \textbf{(Problem 2)}$$

Since costs are non-negative, any policy feasible under (**Problem 2**) is also feasible for the original (**Problem 1**). Setting the standard CMDP expected discounted cost threshold to 0, this re-formulation enables the use of off-the-shelf constrained policy optimisation techniques whilst in principle still enforcing strictly safe behaviour.

**Discussion.** Most prior work focuses on *positive cost thresholds* (e.g. a value of 25 is common in *Safety Gymnasium* settings (Ji et al., 2023)). However, in safety-critical applications there is no meaningful notion of an "acceptable" amount of potentially catastrophic failure. In practice, the level of cost conducive to training progress varies widely across tasks, algorithms, neural network architectures, and optimisation hyperparameters. Using positive cost thresholds in safety-critical contexts makes policy training more brittle, environment-dependent, and often misaligned with true safety objectives. Conceptually, positive thresholds are undesirable in such applications as they conflate the *problem specification* with an *algorithmic hyperparameter*.

## 4 METHOD

We first derive the idealised SB-TRPO update as a relaxation of CPO, then present a practical approximation together with its performance-improvement guarantees, and finally summarise the overall method in a complete algorithmic description.

### 4.1 SAFETY-BIASED TRUST REGION UPDATE

In the setting of hard (zero-cost) constraints, the idealised CPO (Achiam et al., 2017) update seeks a feasibility-preserving improvement of the reward within the trust region:

$$\max_{\pi} \; J_r(\pi) \qquad \text{s.t.} \qquad J_c(\pi) \leq 0, \quad D_{\text{KL}}^{\max}(\pi_{\text{old}} \,\|\, \pi) \leq \delta \qquad \text{(CPO)}$$

This update only considers policies that remain feasible. When no such policy exists within the trust region, CPO (and related methods such as C-TRPO) switches to a recovery step:

$$\min_{\pi} \; J_c(\pi) \qquad \text{s.t.} \quad D_{\text{KL}}^{\max}(\pi_{\text{old}} \,\|\, \pi) \leq \delta \qquad \text{(Recovery)}$$

Empirically, the (Recovery) step does drive the cost down, but often at the expense of extreme conservatism and with little regard for task reward (by design). Once a zero-cost policy has been found, CPO switches back to the feasibility-preserving update (CPO). Starting from these overly cautious policies, any improvement in task reward typically requires temporarily increasing the cost. This is ruled out by the constraint in (CPO). As a consequence, CPO gets "stuck" near zero-cost but task-ineffective policies, never escaping the conservative part of policy space.

To address this limitation, we introduce a more general update rule that seeks high reward while requiring only a controlled reduction of the cost by at least $\epsilon \geq 0$:

$$\max_{\pi} \; J_r(\pi) \qquad \text{s.t.} \qquad J_c(\pi) \leq J_c(\pi_{\text{old}}) - \epsilon, \quad D_{\text{KL}}^{\max}(\pi_{\text{old}} \,\|\, \pi) \leq \delta \qquad \textbf{(Update 1)}$$

Note that CPO corresponds to the special case $\epsilon = J_c(\pi_{\text{old}})$, which forces feasibility at every iteration. In contrast, our formulation does *not* require the intermediate policies to remain feasible (see Section 6 for an extended discussion).

To ensure that (**Update 1**) always admits at least one solution, and to avoid the need for an explicit recovery step such as (Recovery), we choose $\epsilon$ to be a fixed fraction of the best achievable cost improvement inside the trust region. Formally, for a *safety bias* $\beta \in (0, 1]$, we define $\epsilon := \beta \cdot (J_c(\pi_{\text{old}}) - c^*_{\pi_{\text{old}}})$, where

$$c^*_{\pi_{\text{old}}} := \min_{\pi} \; J_c(\pi) \qquad \text{s.t.} \qquad D_{\text{KL}}^{\max}(\pi_{\text{old}} \,\|\, \pi) \leq \delta \qquad (1)$$

Crucially, this guarantees feasibility of (**Update 1**) as well as $\epsilon \geq 0$. The parameter $\beta$ controls how aggressively the algorithm insists on cost reduction at each step.

$\beta = 1$ **recovers CPO.** Setting $\beta = 1$ forces each update to pursue *maximal* cost improvement within the trust region, reducing the cost constraint in (**Update 1**) to $J_c(\pi) \leq c^*_{\pi_{\text{old}}}$. Note that $c^*_{\pi_{\text{old}}} = 0$ iff (CPO) is feasible. Thus, (**Update 1**) for $\beta = 1$ elegantly *captures both the recovery and standard feasible phases of CPO.*

**Intuition for using a safety bias $\beta < 1$.** Choosing $\beta < 1$ intentionally relaxes this requirement: the policy must still reduce cost, but only by a *fraction* of the optimal improvement. This slack provides room for reward-directed updates even when the maximally cost-reducing step would be overly restrictive. In practice, $\beta < 1$ prevents the algorithm from getting trapped in low-reward regions and enables steady progress towards both low cost and high reward.

The theoretical properties of the idealised (**Update 1**) can be summarised as follows:

**Theorem 1.** *Let $\pi_0, \pi_1, \ldots$ be the sequence of policies generated by the idealised (**Update 1**). Then*

1. *The cost is monotonically decreasing: $J_c(\pi_{k+1}) \leq J_c(\pi_k)$ for all $k$.*

2. *Whenever no cost decrease is possible, reward improves:* $J_c(\pi_{k+1}) = J_c(\pi_k)$ *implies* $J_r(\pi_{k+1}) \geq J_r(\pi_k)$.

3. *If for some K neither cost nor reward improves,* $J_c(\pi_{K+1}) = J_c(\pi_K)$ *and* $J_r(\pi_{K+1}) = J_r(\pi_K)$, *then* $\pi_K$ *is a trust-region local optimum of both cost and the modified constrained problem:*

$$J_c(\pi_K) = \min_{\pi : D_{\mathrm{KL}}^{\max}(\pi_K \| \pi) \leq \delta} J_c(\pi) = c^*_{\pi_K} \qquad J_r(\pi_K) = \max_{\substack{\pi : J_c(\pi) \leq c^*_{\pi_K}, \\ D_{\mathrm{KL}}^{\max}(\pi_K \| \pi) \leq \delta}} J_r(\pi)$$

Reward is not necessarily monotonically increasing, and $J_c(\pi_{k+1}) = J_c(\pi_k)$ does not preclude future cost reduction since the trust regions continue to evolve as long as reward improves.

## 4.2 Approximate Solution

Since direct evaluation of $J_r(\pi)$ and $J_c(\pi)$ for arbitrary $\pi$ is infeasible, we employ the standard *surrogate objectives* (Schulman et al., 2015) with respect to a reference policy $\pi_{\mathrm{old}}$. For reward (and analogously for cost), we define

$$\mathcal{L}_{r,\pi_{\mathrm{old}}}(\pi) := J_r(\pi_{\mathrm{old}}) + \mathbb{E}_{s,a \sim \pi_{\mathrm{old}}} \left[ \frac{\pi(a \mid s)}{\pi_{\mathrm{old}}(a \mid s)} \cdot A_r^{\pi_{\mathrm{old}}}(s,a) \right]$$

By the Policy Improvement Bound (Schulman et al., 2015, Thm. 1),

$$\left| \mathcal{L}_{r,\pi_{\mathrm{old}}}(\pi) - J_r(\pi) \right| \leq C_{r,\pi_{\mathrm{old}}} \cdot D_{\mathrm{KL}}^{\max}(\pi_{\mathrm{old}} \| \pi)$$

for a constant $C_{r,\pi_{\mathrm{old}}} \geq 0$. In particular, $J_r(\pi_{\mathrm{old}}) = \mathcal{L}_{r,\pi_{\mathrm{old}}}(\pi_{\mathrm{old}})$, and similar bounds hold for $J_c$. These observations justify approximating the idealised (**Update 1**) by its surrogate:

$$\max_{\pi} \ \mathcal{L}_{r,\pi_{\mathrm{old}}}(\pi) \qquad \text{s.t.} \qquad \mathcal{L}_{c,\pi_{\mathrm{old}}}(\pi) \leq \mathcal{L}_{c,\pi_{\mathrm{old}}}(\pi_{\mathrm{old}}) - \epsilon, \quad D_{\mathrm{KL}}^{\max}(\pi_{\mathrm{old}} \| \pi) \leq \delta \qquad (\textbf{Update 2})$$

As in TRPO, the KL constraint guarantees that the surrogates remain close to the true performance.

**Second-Order Approximation.** Henceforth, we assume differentiably parameterised policies $\pi_\theta$ and overload notation, e.g. $J_c(\theta)$ for $J_c(\pi_\theta)$). Linearising the reward and cost objectives around the current parameters $\theta_{\mathrm{old}}$ and approximating the KL divergence by a quadratic form with Fisher information, yields the quadratic program

$$\max_{\Delta \in \mathbb{R}^d} \langle g_r, \Delta \rangle \quad \text{s.t.} \quad \langle g_c, \Delta \rangle \leq -\epsilon, \quad \tfrac{1}{2} \cdot \Delta^\top \cdot F \cdot \Delta \leq \delta, \qquad (\textbf{Update 3})$$

where $g_r := \nabla \mathcal{L}_{r,\theta_{\mathrm{old}}}(\theta_{\mathrm{old}})$, $g_c := \nabla \mathcal{L}_{c,\theta_{\mathrm{old}}}(\theta_{\mathrm{old}})$ and $F$ denotes the Fisher information matrix.[1] To choose $\epsilon$ and approximate Eq. (1), we analogously approximate the trust-region step that maximally decreases the cost surrogate

$$\Delta_c := \arg\min_{\Delta} \langle g_c, \Delta \rangle \quad \text{s.t.} \quad \tfrac{1}{2} \cdot \Delta^\top \cdot F \cdot \Delta \leq \delta. \qquad (2)$$

and set $\epsilon := -\beta \cdot \langle g_c, \Delta_c \rangle$ with safety-bias $\beta \in (0,1]$. This guarantees feasibility of equation **Update 3**. The new parameters are then updated via $\theta = \theta_{\mathrm{old}} + \Delta^*$, with $\Delta^*$ the solution of (**Update 3**), analogous to the TRPO step but enforcing an explicit local cost reduction.

**Approximate Solution via Cost-Biased Convex Combination.** Solving (**Update 3**) analytically requires computing the natural gradients $F^{-1} g_r$ and $F^{-1} g_c$ and their coefficients (cf. Lemma 4 in Section B.2), which is often numerically fragile for large or ill-conditioned $F$. Instead, we follow TRPO and compute the KL-constrained reward and cost steps separately using the conjugate gradient method (Hestenes et al., 1952):

$$\Delta_r := \arg\max_{\Delta} \langle g_r, \Delta \rangle \quad \text{s.t.} \quad \tfrac{1}{2} \cdot \Delta^\top \cdot F \cdot \Delta \leq \delta \qquad (3)$$

---

[1]The KL constraint corresponds to bounding the *sample-average expected* KL divergence under the current policy, as in TRPO, rather than the maximum KL across states.

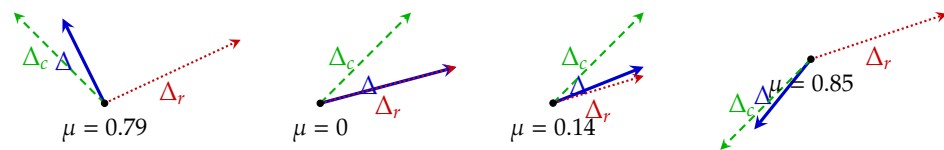

Figure 1: Visualisation of the adaptive convex combination $\Delta$ of $\Delta_r$ and $\Delta_c$ given by Eq. (4) for $\epsilon := 1.4 = -\beta \cdot \langle g_c, \Delta_c \rangle$, where $\beta := 0.7$, and the special case that $\Delta_r = g_r$ and $\Delta_c = -g_c$.

with $\Delta_c$ given by Eq. (2). Both steps satisfy the same KL constraint, and thus any convex combination $\Delta := (1 - \mu) \cdot \Delta_r + \mu \cdot \Delta_c$, where $\mu \in [0, 1]$, also satisfies it (see Lemma 2 in Section B.2). We therefore choose the largest reward-improving combination which also satisfies $\langle g_c, \Delta \rangle \leq -\epsilon$ (see Fig. 1 for a visualisation and Lemma 3):

$$\mu := \begin{cases} \frac{\langle g_c, \Delta_r \rangle + \epsilon}{\langle g_c, \Delta_r \rangle - \langle g_c, \Delta_c \rangle} & \text{if } \langle g_c, \Delta_r \rangle \geq -\epsilon \text{ and } \langle g_c, \Delta_r \rangle \neq \langle g_c, \Delta_c \rangle, \\ 0 & \text{otherwise.} \end{cases} \tag{4}$$

### 4.3 Performance Improvement

This approximation of (**Update 3**) ensures an approximate cost decreases at each update:

$$J_c(\theta + \Delta) - J_c(\theta) \approx \langle g_c, \Delta \rangle \leq \beta \cdot \langle g_c, \Delta_c \rangle \leq 0$$

Likewise, if the reward gradient is non-zero and its angle with $\Delta_c$ does not exceed $90°$, an increase in reward is guaranteed. Formally, sufficiently small steps along $\Delta$ are required for the local gradient approximations to hold:

**Theorem 2** (Performance Improvement). *There exists $\eta \in (0, 1]$ such that*

1. *if $g_c \neq 0$ then $J_c(\theta + \eta \cdot \Delta) < J_c(\theta)$*
2. *if $g_r \neq 0$ and $\langle g_r, \Delta_c \rangle \geq 0$ (in particular, $g_c = 0$) then $J_r(\theta + \eta \cdot \Delta) \geq J_r(\theta)$.*

This result complements Theorem 1 with guarantees for the gradient-based approximate solution of (**Update 1**). Unlike CPO and related methods such as C-TRPO (Milosevic et al., 2024), which have a separate feasibility recovery phase, our update *can guarantee reward improvement for all steps in which the angle between cost and reward gradients does not exceed $90°$*. Once the cost gradient is near zero ($g_c \approx 0$), updates focus on reward improvement (see also Fig. 2a). Overall, this scheme dynamically balances cost and reward objectives, naturally stabilising learning.

### 4.4 Practical Algorithm

Our practical algorithm, implementing the ideas of the preceding subsections, is given in Algorithm 1. We now highlight the key additions.

**Line Search.** In practice, we follow standard TRPO procedures by performing a line search along the cost-biased convex combination $\Delta$: the KL constraint is enforced empirically to ensure that the surrogate approximations of $J_r$ and $J_c$ remain accurate; and the empirical reduction in the surrogate cost loss is checked to operationalise the cost decrease guarantee of Theorem 2, scaling the step if necessary.

**Gradient Estimation.** The gradients of the surrogate objective $g_r = \nabla \mathcal{L}_{r,\pi_{\text{old}}}(\pi_{\text{old}})$ (analogously for cost) coincide with $\nabla J_r(\pi_{\text{old}})$, which is Monte Carlo estimated using the policy gradient theorem. The Fisher information matrix $F$ is similarly estimated from trajectories.

---

**Algorithm 1** Safety-Biased Trust Region Policy Optimisation (SB-TRPO)

---

**Require:** KL divergence limit $\delta > 0$, safety bias $\beta \in [0, 1]$, training epochs $N \in \mathbb{N}$
 1: initialise $\theta$
 2: $\kappa \leftarrow 10^{-8}$ {small constant to avoid division by 0}
 3: **for** $N$ epochs **do**
 4:     $\mathcal{D} \leftarrow$ collect rollouts under policy $\pi_\theta$
 5:     $g_r, g_c \leftarrow$ policy gradient estimate of $\nabla\mathcal{L}_{r,\pi_\theta}(\pi_\theta)$ and $\nabla\mathcal{L}_{c,\pi_\theta}(\pi_\theta)$ using $\mathcal{D}$
 6:     $\Delta_r, \Delta_c \leftarrow$ calculated via the conjugate gradient algorithm to optimise Eqs. (2) and (3)
 7:     $\mu \leftarrow \max\left\{0, \frac{\langle g_c, \Delta_r \rangle - \beta \cdot \langle g_c, \Delta_c \rangle}{\langle g_c, \Delta_r \rangle - \langle g_c, \Delta_c \rangle + \kappa}\right\}$
 8:     $\Delta \leftarrow (1 - \mu) \cdot \Delta_r + \mu \cdot \Delta_c$
 9:     $\eta \leftarrow 1$ {line search for constraint satisfaction}
10:     **repeat**
11:         decrease $\eta$
12:     **until** $\widehat{D_{KL}}[\pi_\theta \parallel \pi_{\theta + \eta \cdot \Delta}] \leq \delta$ and $\widehat{\mathcal{L}_{c,\pi_\theta}}(\pi_{\theta + \eta \cdot \Delta}) \leq \widehat{\mathcal{L}_{c,\pi_\theta}}(\pi_\theta)$
13:     $\theta \leftarrow \theta + \eta \cdot \Delta$
14: **end for**

---

## 5 EMPIRICAL EVALUATION

### 5.1 EXPERIMENTAL SETUP

**Benchmarks.** Safety Gymnasium provides a broad suite of environments for benchmarking safe RL algorithms. Originally introduced by OpenAI (Ray et al., 2019), it has since been maintained and extended by Ji et al. (2023), who also developed *SafePO*, a library of safe RL algorithms.[2] We focus on two classes of environments: *Safe Navigation* and *Safe Velocity*. In Safe Navigation, a robot equipped with LIDAR sensors must complete tasks while avoiding unsafe hazards scattered throughout the environment. We use the *Point* and *Car* robots on four tasks—*Push*, *Button*, *Circle*, and *Goal*—at *level 2*, which is most challenging due to the greatest number of hazards. In contrast, *Safe Velocity* environments test adherence to velocity limits rather than obstacle avoidance, and we use the *Hopper* and *Swimmer* robots adapted from the MuJoCo locomotion suite (Todorov et al., 2012). The state spaces are up to 88-dimensional; see Section C.1.1 for more details and visualisations.

Safety Gymnasium uses reward shaping to provide the agent with auxiliary guidance signals that facilitate more efficient policy optimisation. In the Goal tasks of Safe Navigation environments, for example, the agent receives dense rewards based on its movement relative to the goal, supplementing the sparse task completion reward, so that the cumulative reward can be negative if the agent moves significantly away from the goal. A cost is also computed to quantify constraint violations, with scales that are task dependent. In our selected tasks, these violations include contact with hazards, displacement of hazards, and leaving safe regions in Safe Navigation, and exceeding velocity limits in Safe Velocity.

**Baselines.** We compare our approach with state-of-the-art implementations from SafePO (Ji et al., 2023) of TRPO- and PPO-Lagrangian (Ray et al., 2019), CPO (Achiam et al., 2017), CUP (Yang et al., 2022), CPPO-PID (Stooke et al., 2020), FOCOPS (Zhang et al., 2020), RCPO (Tessler et al., 2018), PCPO (Yang et al., 2020), P3O (Zhang et al., 2022), C-TRPO (Milosevic et al., 2024) and C3PO (Milosevic et al., 2025). These baselines cover the two major model-free constrained RL families: Lagrangian and trust-region/projection-based methods.[3] All baselines require a cost tolerance threshold, which we set to 0 to target almost-sure safety.

---

[2]Some studies customise the environments (e.g., (Jayant & Bhatnagar, 2022; Yu et al., 2022)), for instance by providing more informative observations, making direct comparison of raw performance metrics across works impossible. To ensure fairness and facilitate future comparative studies, we report results only on the standard, unmodified Safety Gymnasium environments (Ji et al., 2023).

[3]Code of reward–cost switching methods such as (P)CRPO (Xu et al., 2021; Gu et al., 2024a) and APPO (Dai et al., 2023) is not publicly available (Gu et al., 2024b).

**Implementation Details.** We use a homoscedastic Gaussian policy with state-independent variance. Following the default setup in SafePO, the policy mean is parameterised by a feedforward network with two hidden layers of 64 units each: the first uses a `tanh` activation, and the second is a linear output layer.

In our implementation of SB-TRPO, we eschew critics both to reduce computational cost and to avoid potential harm from inevitable inaccuracies in critic estimates. Instead, we rely on Monte Carlo estimates, which provide a practical and sufficiently accurate means of enforcing constraints while maximising reward. By contrast, all baseline methods retain their standard critic-based advantage estimation, as this is considered integral.

**Training Details.** We train the policies for $2 \times 10^7$ time steps ($10^3$ epochs), running $2 \times 10^4$ steps per epoch with 20 vectorised environments per seed. All results are reported as averages over 5 seeded parameter initialisations, with standard deviations indicating variability across seeds. Hyperparameter details are presented in Section C.1.2, in particular $\beta \in [0.6, 0.9]$ (see also Section 5.3 for an ablation study). Training is performed on a server with an NVIDIA H200 GPU.

**Metrics.** We compare our performance with the baselines using the standard metrics of *rewards* and *costs*, averaged over the past 50 episodes during training. End-of-training values provide a quantitative comparison, while full training curves reveal the asymptotic behaviour of reward and cost as training approaches the budgeted limit.

To capture safety under hard constraints, we introduce three additional metrics. *Safety probability* is the fraction of episodes completed without any safety violations, i.e., with zero cost. *Safe rewards* denote the average return over episodes completed without violations, so that only safe episodes contribute. Finally, the *safety-biased–cost–reward (SCR) metric* combines both with cost, defined as

$$\text{SCR} := \frac{\text{safety probability}}{\text{cost} + 1} \times \text{safe rewards}$$

SCR integrates complementary dimensions of performance into a single measure: it rewards methods that complete a higher fraction of fully safe episodes (capturing the strict notion of safety), penalises larger costs (capturing extent of violations, cumulatively), and accounts for rewards achieved in those safe episodes (capturing task performance under safety). This balance ensures that higher SCR values correspond to policies that achieve meaningful task completion while adhering closely to safety constraints.

## 5.2 RESULTS

The results are shown in Tables 1 and 2. Best performance in cost and safety probability does not always coincide (similarly for (safe) rewards). PPO-Lagrangian often collapses to poor reward, poor safety, or both. Baselines such as TRPO-Lagrangian, CPO or C-TRPO can achieve better feasibility on some harder tasks (e.g. Point Button), but their rewards are very low—mostly negative—yielding minimal task completion and still falling far short of almost-sure safety. CUP, CPPO-PID and FOCOPS generally have larger constraint violations (except for Swimmer Velocity).

Our method is optimal for at least one metric across tasks and hence consistently Pareto optimal: no other algorithm achieves lower cost without reducing reward, or higher reward without increasing cost (and similarly for safety probability vs safe reward). Furthermore, among algorithms achieving positive rewards, we always attain the best cost or safety probability. Even in exceptions—Hopper Velocity, where we achieve an excellent 99% safety with strong (safe) rewards, and Car Push, where C3PO attains slightly lower cost but barely positive reward—SB-TRPO maintains a clear practical advantage, reliably balancing both high performance and high safety.

Our method attains the best safety-biased–cost–reward (SCR) metric (cf. Table 2) on all tasks except for Swimmer Velocity. Besides, eschewing critics, we beat baselines by at least a factor of 20 in computational cost per epoch (see Table 8 in Section C.2).

Table 1: Rewards and Costs, as well as safety probability and safe reward across 8 benchmark tasks (all at level 2). Red highlights no meaningful task completion (negative (safe) reward) and grey data pairs are not pareto-optimal. Furthermore, we bold the individual best results in each category. ↑ indicates higher values are better and vice versa.

| Method | Point Push | | Point Button | | Point Goal | | Car Push | | Car Circle | | Car Goal | | Hopper Velocity | | Swimmer Velocity | |
|---|---|---|---|---|---|---|---|---|---|---|---|---|---|---|---|---|
| | Rewards↑ | Costs↓ | Rewards↑ | Costs↓ | Rewards↑ | Costs↓ | Rewards↑ | Costs↓ | Rewards↑ | Costs↓ | Rewards↑ | Costs↓ | Rewards↑ | Costs↓ | Rewards↑ | Costs↓ |
| Ours | 0.53 ± 0.19 | 9.2 ± 5.9 | 1.9 ± 0.7 | 15 ± 5 | 1.5 ± 0.6 | 16 ± 7 | 0.55 ± 0.16 | 16 ± 8 | 7.8 ± 2.1 | 0.72 ± 2.80 | 1.1 ± 0.5 | 16 ± 6 | 900 ± 248 | 0.0063 ± 0.0127 | 49 ± 27 | 0.87 ± 3.24 |
| PPO-Lag | −0.47 ± 1.31 | 27 ± 21 | 3.2 ± 1.2 | 76 ± 27 | 2.8 ± 2.2 | 42 ± 18 | −0.20 ± 0.52 | 34 ± 20 | 8.7 ± 2.0 | 22 ± 37 | 2.6 ± 2.5 | 54 ± 29 | 890 ± 507 | 2.9 ± 2.8 | 40 ± 60 | 6.5 ± 7.2 |
| TRPO-Lag | −0.65 ± 0.55 | 9.3 ± 14.5 | −0.72 ± 0.79 | 15 ± 6 | 0.11 ± 0.35 | 17 ± 9 | −0.21 ± 0.27 | 6.3 ± 4.6 | **13 ± 1** | 18 ± 33 | 0.29 ± 0.50 | 16 ± 7 | 160 ± 77 | 2.1 ± 2.6 | 21 ± 16 | 6.3 ± 6.0 |
| CPO | −1.6 ± 1.1 | 5.8 ± 9.6 | −3.2 ± 1.2 | 5.8 ± 2.8 | −1.5 ± 0.5 | 9.3 ± 8.0 | −1.4 ± 0.3 | 4.2 ± 4.4 | 0.96 ± 0.75 | 0.98 ± 6.63 | −0.89 ± 0.27 | 13 ± 8 | 880 ± 185 | **0.0 ± 0.0** | −3.8 ± 9.4 | 0.016 ± 0.037 |
| CUP | 0.18 ± 0.26 | 25 ± 22 | 5.2 ± 1.1 | 120 ± 19 | **7.2 ± 2.8** | 110 ± 28 | −0.026 ± 0.362 | 85 ± 50 | 9.6 ± 2.5 | 48 ± 86 | 6.1 ± 2.8 | 110 ± 31 | 1400 ± 114 | 0.43 ± 0.35 | 38 ± 58 | 0.14 ± 0.18 |
| CPPO-PID | −3.4 ± 3.3 | 13 ± 17 | −1.8 ± 0.8 | 24 ± 9 | −2.3 ± 0.9 | 17 ± 10 | −0.99 ± 0.78 | 34 ± 28 | 35 ± 24 | 27 ± 55 | −0.79 ± 0.69 | 23 ± 10 | 1000 ± 228 | 0.081 ± 0.107 | 0.71 ± 10.89 | 0.044 ± 0.081 |
| FOCOPS | −0.29 ± 0.76 | 35 ± 61 | **9.9 ± 2.0** | 88 ± 14 | 6.5 ± 3.3 | 88 ± 37 | −0.00026 ± 0.41018 | 12 ± 7 | 8.4 ± 1.9 | 39 ± 18.2 | 6.0 ± 5.3 | 86 ± 49 | 970 ± 309 | 0.093 ± 0.135 | 20 ± 12 | 3.0 ± 5.7 |
| RCPO | −0.57 ± 0.51 | 6.3 ± 8.8 | −0.72 ± 0.79 | 15 ± 6 | −0.00008 ± 0.41018 | 12 ± 7 | −0.60 ± 0.49 | 19 ± 20 | 11 ± 2 | 13 ± 35 | 0.29 ± 0.50 | 16 ± 7 | 470 ± 361 | 0.97 ± 1.93 | 19 ± 18 | 5.1 ± 6.7 |
| PCPO | −1.4 ± 0.8 | 14 ± 23 | −2.2 ± 0.8 | 8.8 ± 4.7 | −1.6 ± 0.7 | 18 ± 13 | −0.58 ± 0.46 | 17 ± 18 | 12 ± 1 | 4.4 ± 8.5 | −0.95 ± 0.30 | 17 ± 9 | 750 ± 589 | 3.9 ± 4.9 | −18 ± 2 | 70 ± 16 |
| P3O | 0.3 ± 0.1 | 15 ± 11 | −0.0073 ± 0.2670 | 23 ± 8 | 0.05 ± 0.77 | 36 ± 18 | 0.38 ± 0.15 | 28 ± 12 | 7.6 ± 2.3 | 11 ± 17 | 0.73 ± 0.50 | 40 ± 17 | **1700 ± 34** | 8.9 ± 11.7 | **110 ± 58** | 0.22 ± 0.16 |
| C-TRPO | −1.7 ± 1.4 | 7.8 ± 12.2 | −2.3 ± 0.2 | 6.0 ± 3.0 | −2.5 ± 1.3 | 8.7 ± 5.9 | −1.3 ± 0.5 | 8.9 ± 11.5 | 0.56 ± 0.48 | 3.9 ± 7.4 | −2.1 ± 0.6 | 8.5 ± 6.7 | 6.9 ± 3.5 | 0.0 ± 0.0 | −18 ± 8 | 0.058 ± 0.180 |
| C3PO | −0.18 ± 0.21 | 6.3 ± 5.4 | −0.39 ± 0.16 | 7.8 ± 3.5 | −0.056 ± 0.0720 | 7.3 ± 5.4 | 0.078 ± 0.160 | 12 ± 6 | 5.1 ± 1.7 | 0.53 ± 1.00 | −0.28 ± 0.44 | 12 ± 10 | 1600 ± 45 | 0.48 ± 0.41 | 85 ± 60 | 0.10 ± 0.13 |

| Method | Point Push | | Point Button | | Point Goal | | Car Push | | Car Circle | | Car Goal | | Hopper Velocity | | Swimmer Velocity | |
|---|---|---|---|---|---|---|---|---|---|---|---|---|---|---|---|---|
| | Safe Rew.↑ | Safe Prob.↑ | Safe Rew.↑ | Safe Prob.↑ | Safe Rew.↑ | Safe Prob.↑ | Safe Rew.↑ | Safe Prob.↑ | Safe Rew.↑ | Safe Prob.↑ | Safe Rew.↑ | Safe Prob.↑ | Safe Rew.↑ | Safe Prob.↑ | Safe Rew.↑ | Safe Prob.↑ |
| Ours | 0.53 ± 0.20 | 0.73 ± 0.07 | 1.9 ± 0.8 | 0.47 ± 0.10 | 1.5 ± 0.6 | 0.72 ± 0.08 | 0.55 ± 0.17 | 0.71 ± 0.08 | 7.8 ± 2.1 | 0.99 ± 0.04 | 1.1 ± 0.5 | 0.74 ± 0.07 | 900 ± 247 | 0.99 ± 0.01 | 49 ± 27 | **0.98 ± 0.02** |
| PPO-Lag | −0.49 ± 1.34 | 0.59 ± 0.22 | 3.2 ± 1.3 | 0.18 ± 0.11 | 2.8 ± 2.2 | 0.47 ± 0.13 | −0.20 ± 0.53 | 0.65 ± 0.10 | 8.7 ± 2.2 | 0.86 ± 0.17 | 2.6 ± 2.5 | 0.45 ± 0.16 | 870 ± 542 | 0.58 ± 0.44 | 32 ± 62 | 0.57 ± 0.42 |
| TRPO-Lag | −0.63 ± 0.57 | 0.78 ± 0.16 | −0.71 ± 0.79 | 0.62 ± 0.10 | 0.11 ± 0.36 | **0.79 ± 0.06** | −0.21 ± 0.28 | 0.89 ± 0.04 | **13 ± 1** | 0.81 ± 0.26 | 0.31 ± 0.51 | 0.75 ± 0.07 | 120 ± 115 | 0.6 ± 0.5 | 4.0 ± 14.5 | 0.33 ± 0.42 |
| CPO | −1.6 ± 1.1 | 0.77 ± 0.19 | −3.2 ± 1.2 | 0.76 ± 0.08 | −1.5 ± 0.5 | 0.86 ± 0.05 | −1.4 ± 0.3 | 0.93 ± 0.05 | 0.96 ± 0.76 | 0.99 ± 0.03 | −0.89 ± 0.27 | 0.82 ± 0.09 | 880 ± 182 | **1.0 ± 0.0** | −3.9 ± 9.4 | 0.99 ± 0.03 |
| CUP | 0.18 ± 0.27 | 0.67 ± 0.19 | 2.4 ± 2.3 | 0.033 ± 0.043 | 6.5 ± 2.9 | 0.15 ± 0.09 | −0.013 ± 0.370 | 0.47 ± 0.19 | 9.4 ± 3.0 | 0.75 ± 0.28 | 6.0 ± 2.9 | 0.15 ± 0.10 | 1400 ± 114 | 0.72 ± 0.19 | 38 ± 58 | 0.90 ± 0.10 |
| CPPO-PID | −3.4 ± 3.4 | 0.77 ± 0.20 | −1.3 ± 0.6 | 0.61 ± 0.10 | −1.7 ± 0.3 | 0.78 ± 0.09 | −0.98 ± 0.79 | 0.69 ± 0.15 | 1.5 ± 1.7 | 0.89 ± 0.16 | −0.72 ± 0.58 | 0.73 ± 0.10 | 1000 ± 253 | 0.81 ± 0.21 | −1.5 ± 11.4 | 0.88 ± 0.09 |
| FOCOPS | −0.28 ± 0.78 | 0.67 ± 0.22 | **5.8 ± 3.8** | 0.11 ± 0.05 | **6.8 ± 2.9** | 0.17 ± 0.17 | 0.069 ± 0.351 | 0.67 ± 0.13 | 8.5 ± 1.8 | 0.92 ± 0.14 | **6.1 ± 5.0** | 0.071 ± 0.257 | 960 ± 334 | 0.82 ± 0.22 | 5.2 ± 14.8 | 0.52 ± 0.19 |
| RCPO | −0.57 ± 0.54 | 0.75 ± 0.17 | −0.96 ± 0.94 | 0.64 ± 0.09 | −0.27 ± 0.78 | 0.80 ± 0.07 | −0.60 ± 0.49 | 0.78 ± 0.11 | 11 ± 2 | 0.86 ± 0.21 | 0.29 ± 0.51 | **0.75 ± 0.08** | 450 ± 388 | 0.74 ± 0.28 | 8.4 ± 20.0 | 0.61 ± 0.18 |
| PCPO | −1.4 ± 0.8 | 0.78 ± 0.19 | −1.9 ± 0.6 | 0.70 ± 0.10 | −1.2 ± 0.3 | 0.78 ± 0.09 | −0.58 ± 0.46 | 0.8 ± 0.1 | 12 ± 2 | 0.89 ± 0.14 | −0.91 ± 0.27 | 0.78 ± 0.09 | 720 ± 615 | 0.71 ± 0.31 | −15 ± −4 | 0.89 ± 0.00 |
| P3O | 0.3 ± 0.2 | 0.71 ± 0.14 | −0.19 ± 0.58 | 0.60 ± 0.09 | 0.094 ± 0.736 | 0.61 ± 0.10 | 0.38 ± 0.17 | 0.72 ± 0.08 | 7.3 ± 2.6 | 0.91 ± 0.10 | 0.70 ± 0.56 | 0.59 ± 0.11 | **1700 ± 41** | 0.43 ± 0.46 | **110 ± 58** | 0.91 ± 0.09 |
| C-TRPO | −1.7 ± 1.4 | 0.78 ± 0.15 | −1.7 ± 0.1 | 0.72 ± 0.08 | −1.6 ± 0.3 | 0.84 ± 0.07 | −1.3 ± 0.5 | 0.86 ± 0.07 | 0.69 ± 0.45 | 0.95 ± 0.07 | −2.0 ± 0.4 | 0.83 ± 0.08 | −40 ± 24 | 0.93 ± 0.05 | −18 ± 8 | 0.91 ± 0.09 |
| C3PO | −0.17 ± 0.21 | **0.85 ± 0.05** | −0.39 ± 0.18 | 0.76 ± 0.07 | −0.046 ± 0.735 | 0.86 ± 0.06 | 0.077 ± 0.168 | **0.86 ± 0.05** | 5.1 ± 1.7 | 0.99 ± 0.01 | −0.29 ± 0.45 | 0.86 ± 0.06 | 1600 ± 47 | 0.79 ± 0.14 | 85 ± 60 | 0.92 ± 0.08 |

Table 2: Safety-biased–cost–reward (SCR) metric across 8 benchmark tasks.

| Method | Point Push | Point Button | Point Goal | Car Push | Car Circle | Car Goal | Hopper Velocity | Swimmer Velocity |
|---|---|---|---|---|---|---|---|---|
| Ours | **0.037** | **0.058** | **0.064** | **0.023** | **4.5** | **0.047** | **890** | 26 |
| PPO-Lag | -0.010 | 0.0075 | 0.031 | -0.0037 | 0.33 | 0.021 | 130 | 2.4 |
| TRPO-Lag | -0.048 | -0.027 | 0.0051 | -0.025 | 0.57 | 0.013 | 23 | 0.18 |
| CPO | -0.18 | -0.36 | -0.12 | -0.25 | 0.48 | -0.051 | 880 | -3.7 |
| CUP | 0.0047 | 0.00064 | 0.0085 | -0.000071 | 0.14 | 0.0087 | 720 | 30 |
| CPPO-PID | -0.19 | -0.032 | -0.077 | -0.019 | 0.048 | -0.022 | 770 | -1.3 |
| FOCOPS | -0.0053 | 0.0069 | 0.013 | 0.0013 | 1.6 | 0.005 | 720 | 0.67 |
| RCPO | -0.058 | -0.038 | -0.017 | -0.023 | 0.71 | 0.013 | 170 | 0.84 |
| PCPO | -0.075 | -0.14 | -0.049 | -0.026 | 2.0 | -0.039 | 100 | -0.19 |
| P3O | 0.013 | -0.0048 | 0.0015 | 0.0093 | 0.53 | 0.010 | 75 | **81** |
| C-TRPO | -0.15 | -0.18 | -0.14 | -0.11 | 0.13 | -0.17 | -37 | -15 |
| C3PO | -0.020 | -0.034 | -0.0048 | 0.005 | 3.3 | -0.02 | 870 | 71 |

Our approach also exhibits robustness in practice: temporary increases in cost are typically corrected quickly whilst improving rewards overall (see Fig. 2a). Besides, Figs. 4 and 5 in Section C.2 show that longer training can continue to improve both reward and feasibility for our method, while the baselines plateau.

In summary, our approach is the only one to consistently achieve the *best balance of safety and meaningful task completion*.

### 5.3 Ablation Studies

We ablate key design choices to assess their impact on performance.

**Safety Bias $\beta$.** We evaluate the effect of the safety bias $\beta$ over $\beta \in [0.6, 0.9]$ on Point Button and Car Goal. Fig. 2c shows that varying $\beta$ shifts SB-TRPO policies along a nearly linear reward-cost Pareto frontier: lower values emphasise safety at the expense of reward, while higher values trade off some safety for greater reward. By contrast, the baselines typically underperform relative to this frontier (or achieve negative reward). This demonstrates that SB-TRPO is robust to the choice of $\beta$ (with $\beta \approx 0.7$ serving as an excellent starting point throughout) and yields higher rewards for the same level of constraint violation.

**Eschewing Critics.** To further justify our design choice of omitting critics (see Implementation Details in Section 5.1), we ablate the effect of adding a critic to our SB-TRPO method in Car Goal and Point Button (Fig. 2b). Including a critic boosts reward learning but also increases cost, highlighting the tradeoff between reward optimisation and safety. Our main implementation omits the critic, yielding more stable and conservative policies that better satisfy safety constraints, whilst reducing the computational burden significantly (see Table 8). Even with the higher cost, the critic variant still outperforms baselines on the SCR metric (cf. Tables 2 and 9).

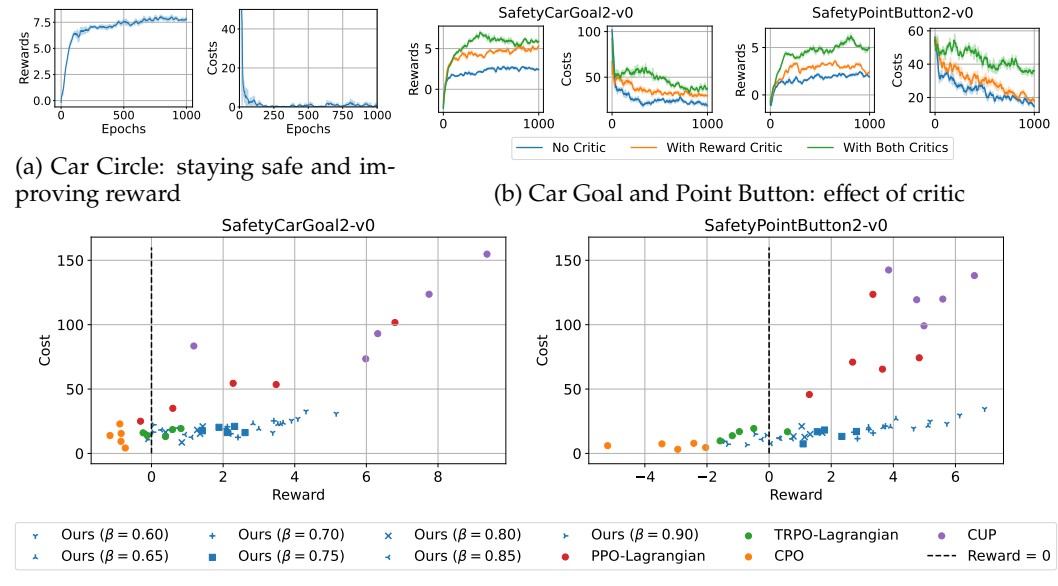

(a) Car Circle: staying safe and improving reward

(b) Car Goal and Point Button: effect of critic

(c) Ablation study of safety bias $\beta$

Figure 2: Maintaining safety and ablation studies.

## 6 Discussion

CPO and related methods like C-TRPO enforce expected cost constraints at every step, resorting to purely cost-focused recovery when safety violations occur. This slows learning and pushes the policy into *trivially safe* regions (e.g. corners without hazards), from which escape would require temporary violations that CPO forbids, causing stagnation. Moreover, reward improvement is *not guaranteed* during these feasibility-recovery phases.

Our empirical results demonstrate that framing policy updates as (**Update 3**), focusing on cost reduction rather than strict feasibility in each step, effectively prevents collapse into trivially safe but task-ineffective regions. As illustrated in Fig. 7, SB-TRPO's updates, which dynamically balance reward and cost, align well with both gradients, whereas CPO updates are often at best orthogonal to reward. By Theorem 2, this alignment *guarantees consistent progress in both safety and task performance*. Furthermore, by *not switching between separate phases for reward and feasibility recovery*, learning is markedly *smoother*.

Lagrangian methods suffer from monotonically increasing multipliers in the zero-cost setting, making performance highly sensitive to their initial value: small values risk unsafe policies, large ones induce over-conservatism, and once increased the multiplier cannot decrease, often trapping the policy in conservative regimes. CUP, which uses a similar primal–dual mechanism, shows analogous limitations.

**Conclusion.** SB-TRPO introduces a principled trust-region update that ensures progress towards safety while simultaneously seeking reward improvement. This produces *smoother learning*, prevents collapse into overly conservative solutions, and enables *steady reward accumulation*. Across *Safety Gymnasium* tasks, SB-TRPO consistently outperforms state-of-the-art methods in balancing high safety with task performance.

**Limitations.** SB-TRPO targets hard contraints: it is not directly applicable to CMDPs with positive cost thresholds (although (**Update 3**) can replace the conventional recovery stage of e.g. CPO). As with other policy optimisation methods (Schulman et al., 2015; Gu et al., 2024a), our performance guarantee (Theorem 2) assumes exact gradients and holds only approximately with estimates. Moreover, while our method achieves strong results in *Safety Gymnasium*, it does not guarantee almost-sure safety on the most challenging benchmarks.

ETHICS STATEMENT

This is foundational research on reinforcement learning with safety constraints, studied entirely in simulation. While our methods aim to reduce unsafe behaviour, they are not sufficient for direct deployment in safety-critical domains. We see no negative societal impacts and expect this work to contribute to safer foundations for future reinforcement learning systems.

REPRODUCIBILITY STATEMENT

Code and scripts to reproduce all results are provided in the supplementary materials (see Section C.1.3). Implementation details and hyperparameters are described in Sections C.1.2 and 5.1, while environment specifications and ablation studies are given in Sections C.1.1 and 5.1, and Sections C.3 and 5.3, respectively. Proofs of the theoretical results of Section 4 are provided in Section B.

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

## A    Motivation of Zero-Cost CMDP Formulation

In many safety-critical domains, the appropriate requirement is that the agent *never* enters unsafe states (e.g. when they encode damage to an expensive robot or even accidents of autonomous cars). In the discounted-cost CMDP formalism with non-negative costs, this requirement is expressed *exactly* by imposing the zero-threshold constraint $J_c(\pi) = 0$. Introducing a positive threshold $b > 0$ effectively permits some violations and, crucially, adds a hyperparameter that encodes an arbitrary notion of "acceptable risk".

The level of violation required for learning to progress varies greatly across tasks (e.g. around 17 on the hardest navigation environments, around 10 for PointPush, and well below 1 for velocity or circle tasks) as well as across algorithms, neural network architectures and optimisation hyperparameters. This makes positive-threshold CMDPs brittle, environment-dependent, and potentially misaligned with true safety objectives. Conceptually, a positive cost threshold in the context of hard safety is undesirable because it conflates the *problem statement/specification* with an *algorithmic hyper-parameter*.

For these reasons, we focus on the *intrinsically meaningful* zero-threshold (hard-constraint) CMDP, which avoids the ambiguity of selecting $b$, removes the tuning burden associated with calibrating allowable violations, and directly matches the requirement of "almost surely no unsafe events".

Finally, we believe that the zero-cost problem is significantly under-explored in model-free Safe RL, despite being the more appropriate formulation for capturing critical safety violations in practice. By explicitly targeting this regime, our work aims to help shift attention towards this practically important but comparatively understudied problem class.

## B    Supplementary Materials for Section 4

### B.1    Supplementary Materials for Section 4.1

**Lemma 1.** $\mathcal{L}_{r,\pi_{\mathrm{old}}}(\pi) - C_{r,\pi_{\mathrm{old}}} \cdot D_{\mathrm{KL}}^{\max}(\pi_{\mathrm{old}} \parallel \pi) \leq J_r(\pi) \leq \mathcal{L}_{r,\pi_{\mathrm{old}}}(\pi) + C_{r,\pi_{\mathrm{old}}} \cdot D_{\mathrm{KL}}^{\max}(\pi_{\mathrm{old}} \parallel \pi).$

*Proof.* By (Schulman et al., 2015, Thm. 1),

$$\mathcal{L}_{r,\pi_{\mathrm{old}}}(\pi) - C_{r,\pi_{\mathrm{old}}} \cdot D_{\mathrm{KL}}^{\max}(\pi_{\mathrm{old}} \parallel \pi) \leq J_r(\pi), \qquad C_{r,\pi_{\mathrm{old}}} := \frac{4 \cdot \gamma \cdot \max_{s,a} |A_r^{\pi_{\mathrm{old}}}(s,a)|}{(1 - \gamma)^2}$$

Consider the reward function $-r$. Note that $J_{-r}(\pi) = -J_r(\pi)$ and $A_{-r}^{\pi_{\mathrm{old}}}(s,a) = -A_r^{\pi_{\mathrm{old}}}(s,a)$. Thus, $C_{-r,\pi_{\mathrm{old}}} = C_{r,\pi_{\mathrm{old}}}$ and $\mathcal{L}_{-r,\pi_{\mathrm{old}}}(\pi) = -\mathcal{L}_{r,\pi_{\mathrm{old}}}(\pi)$. Finally, again by (Schulman et al., 2015, Thm. 1),

$$-J_r(\pi) = J_{-r}(\pi) \geq \mathcal{L}_{-r,\pi_{\mathrm{old}}}(\pi) - C_{-r,\pi_{\mathrm{old}}} \cdot D_{\mathrm{KL}}^{\max}(\pi_{\mathrm{old}} \parallel \pi)$$
$$= -\mathcal{L}_{r,\pi_{\mathrm{old}}}(\pi) - C_{r,\pi_{\mathrm{old}}} \cdot D_{\mathrm{KL}}^{\max}(\pi_{\mathrm{old}} \parallel \pi) \qquad \square$$

**Theorem 1.** *Let $\pi_0, \pi_1, \ldots$ be the sequence of policies generated by the idealised (**Update 1**). Then*

1. *The cost is monotonically decreasing: $J_c(\pi_{k+1}) \leq J_c(\pi_k)$ for all $k$.*
2. *Whenever no cost decrease is possible, reward improves: $J_c(\pi_{k+1}) = J_c(\pi_k)$ implies $J_r(\pi_{k+1}) \geq J_r(\pi_k)$.*
3. *If for some K neither cost nor reward improves, $J_c(\pi_{K+1}) = J_c(\pi_K)$ and $J_r(\pi_{K+1}) = J_r(\pi_K)$, then $\pi_K$ is a trust-region local optimum of both cost and the modified constrained problem:*

$$J_c(\pi_K) = \min_{\pi: D_{\mathrm{KL}}^{\max}(\pi_K \| \pi) \leq \delta} J_c(\pi) = c_{\pi_K}^* \qquad J_r(\pi_K) = \max_{\substack{\pi: J_c(\pi) \leq c_{\pi_K}^*, \\ D_{\mathrm{KL}}^{\max}(\pi_K \| \pi) \leq \delta}} J_r(\pi)$$

*Proof sketch.* Note that $J_c(\pi_{k+1}) = J_c(\pi_k)$ implies $\epsilon = 0$. The theorem follows directly by definition of (**Update 1**). $\qquad \square$

## B.2 Supplementary Materials for Section 4.2

We recall the Fisher information matrix:

$$F := \mathbb{E}_{s,a \sim \pi_{\theta_{\text{old}}}} \left[ \nabla_\theta \log \pi_\theta(a|s) \big|_{\theta=\theta_{\text{old}}} \cdot \nabla_\theta \log \pi_\theta(a|s)^\top \big|_{\theta=\theta_{\text{old}}} \right]$$

**Lemma 2.** *For $\Delta_r$ and $\Delta_c$ defined in Eqs. (2) and (3),*

$$\frac{1}{2} \cdot \Delta^T \cdot F \cdot \Delta \leq \delta$$

*where $\Delta = (1-\mu) \cdot \Delta_r + \mu \cdot \Delta_c$ for arbitrary $\mu \in [0,1]$*

*Proof.* Since $F$ is positive semi-definite, $\Delta \mapsto \Delta^T \cdot F \cdot \Delta$ is convex and the bound follows since it is satisfied by both $\Delta_r$ and $\Delta_c$. □

**Lemma 3.** *Assume $\langle g_c, \Delta_c \rangle \leq -\epsilon$ and define $\Delta_\mu := (1-\mu) \cdot \Delta_r + \mu \cdot \Delta_c$. Then*

$$\max_\mu \langle g_r, \Delta_\mu \rangle \qquad s.t. \quad \langle g_c, \Delta_\mu \rangle \leq -\epsilon, \quad \mu \in [0,1]$$

*is feasible and its optimal solution is $\mu$ defined in Eq. (4).*

NB for our choice $\epsilon := -\beta \cdot \langle g_c, \Delta_c \rangle \geq 0$, where $\beta \in (0,1]$, the assumption clearly holds.

*Proof.* By definition of $\Delta_r$ and $\Delta_c$,

$$\langle g_r, \Delta_r \rangle \geq \langle g_r, \Delta_c \rangle \qquad\qquad \langle g_c, \Delta_c \rangle \leq \langle g_c, \Delta_r \rangle \qquad (5)$$

Therefore, we can equivalently optimise

$$\min_\mu \mu \qquad s.t. \quad \langle g_c, \Delta_\mu \rangle \leq -\epsilon, \quad \mu \in [0,1]$$

If $\langle g_c, \Delta_r \rangle = \langle g_c, \Delta_c \rangle$ or $\langle g_c, \Delta_r \rangle < -\epsilon$ then the optimal solution is clearly $\mu := 0$.

Otherwise, first note that by the premise of this lemma and Eq. (5), the ratio in Eq. (4) is always in $[0,1]$. Besides, it is straightforward to see that $\langle g_c, \Delta_\mu \rangle \leq -\epsilon$ iff

$$\mu \geq \frac{\epsilon + \langle g_c, \Delta_r \rangle}{\langle g_c, \Delta_r \rangle - \langle g_c, \Delta_c \rangle} \qquad\qquad\qquad □$$

**Lemma 4** (Exact solution of (**Update 3**)). *Let $g_r, g_c \in \mathbb{R}^d$, $F \in \mathbb{R}^{d \times d}$ and let $\epsilon > 0$, $\delta > 0$. We assume the* non-degenerate case:

$$g_r \neq 0, \quad g_c \neq 0, \quad g_c \nparallel g_r, \quad F > 0.$$

*Then the optimal solution $\Delta^*$ of (**Update 3**) is*

$$\Delta^* = \begin{cases} \sqrt{\frac{2\delta}{a}} \, F^{-1} g_r, & \text{if } \sqrt{\frac{2\delta}{a}} \, b \leq -\epsilon, \\ \sqrt{\frac{2\delta}{a - 2\lambda^* b + (\lambda^*)^2 c}} \, F^{-1}(g_r - \lambda^* g_c), & \text{otherwise,} \end{cases}$$

*where*

$$a := \langle g_r, F^{-1} g_r \rangle, \quad b := \langle g_r, F^{-1} g_c \rangle, \quad c := \langle g_c, F^{-1} g_c \rangle.$$

*and $\lambda^* > 0$ is the unique positive solution of the quadratic*

$$A\lambda^2 + B\lambda + C = 0, \quad with \quad A = 2\delta c^2 - \epsilon^2 c, \quad B = -4\delta bc + 2\epsilon^2 b, \quad C = 2\delta b^2 - \epsilon^2 a.$$

*Proof.* We form the Lagrangian with multipliers $\lambda \geq 0$ (linear constraint) and $\nu \geq 0$ (quadratic constraint):

$$\mathcal{L}(\Delta, \lambda, \nu) = \langle g_r, \Delta \rangle + \lambda(-\epsilon - \langle g_c, \Delta \rangle) + \nu \left( \delta - \tfrac{1}{2} \Delta^\top F \Delta \right).$$

Stationarity gives

$$g_r - \lambda g_c - \nu F \Delta = 0 \implies \Delta = \frac{1}{\nu} F^{-1}(g_r - \lambda g_c), \quad \nu > 0.$$

**Justification that $\nu > 0$.** The objective is linear in $\Delta$ and $g_r \neq 0$, so it is unbounded in the direction of $g_r$ without the quadratic constraint. The linear constraint $\langle g_c, \Delta \rangle \leq -\epsilon$ defines a half-space. In the non-degenerate case $g_c \nparallel g_r$, this half-space alone cannot bound the linear objective. Therefore, the quadratic constraint must be active at the solution, which by complementary slackness implies $\nu > 0$.

**Form of KKT candidates.** Substituting $\Delta$ into the quadratic constraint $\frac{1}{2}\Delta^\top F \Delta = \delta$ gives

$$\nu = \sqrt{\frac{(g_r - \lambda g_c)^\top F^{-1}(g_r - \lambda g_c)}{2\delta}}.$$

Hence all KKT candidates satisfy

$$\Delta(\lambda) = \sqrt{\frac{2\delta}{(g_r - \lambda g_c)^\top F^{-1}(g_r - \lambda g_c)}} \cdot F^{-1} \cdot (g_r - \lambda g_c)$$

$$= \sqrt{\frac{2\delta}{a - 2\lambda b + \lambda^2 c}} \cdot F^{-1} \cdot (g_r - \lambda g_c)$$

using the above definitions of $a$, $b$ and $c$, so that

$$\langle g_c, \Delta(\lambda) \rangle = \sqrt{\frac{2\delta}{a - 2\lambda b + \lambda^2 c}} \cdot (b - \lambda c).$$

**Complementary slackness for the linear constraint.** Either $\lambda = 0$ (linear constraint inactive) or $\lambda > 0$ with equality $\langle g_c, \Delta(\lambda) \rangle = -\epsilon$.

**Case $\lambda = 0$.** If $\langle g_c, \Delta(0) \rangle = \sqrt{\frac{2\delta}{a}}\, b \leq -\epsilon$, then $\Delta^* = \Delta(0)$.

**Case $\lambda > 0$.** Solving $\langle g_c, \Delta(\lambda) \rangle = -\epsilon$ and squaring both sides gives the quadratic equation $A\lambda^2 + B\lambda + C = 0$ displayed in the proposition. The unique positive root $\lambda^*$ yields the optimal step $\Delta^* = \Delta(\lambda^*)$.

**Summary.** In the non-degenerate case, the quadratic constraint is active ($\nu > 0$) to ensure a finite optimum. The linear constraint determines $\lambda^*$, and $\Delta^*$ follows directly. □

### B.3 Supplementary Materials for Section 4.3

For $M > 0$, let $B_M(\theta)$ be the $M$-ball around $\theta$.

**Lemma 5.** *Assume $J_r$ and $J_c$ are L-Lipschitz smooth on $B_M(\theta)$ and $\|\Delta_r\|, \|\Delta_c\| \leq M$ for some $M > 0$. Then for all $\eta \in [0, 1]$,*

1. *$J_c(\theta + \eta \cdot \Delta) \leq J_c(\theta) + \eta \cdot \beta \cdot \langle g_c, \Delta_c \rangle + \frac{L \cdot \eta^2}{2} \cdot \|\Delta\|^2$*
2. *$J_r(\theta + \eta \cdot \Delta) \geq J_r(\theta) + \eta \cdot (1 - \mu) \cdot \langle g_r, \Delta_c \rangle + \eta \cdot \mu \cdot \langle g_r, \Delta_r \rangle - \frac{L \cdot \eta^2}{2} \cdot \|\Delta\|^2$*
3. *if $g_c = 0$ then $J_r(\theta + \eta \cdot \Delta) \geq J_r(\theta) + \eta \cdot \langle g_r, \Delta_r \rangle - \frac{L \cdot \eta^2}{2} \cdot \|\Delta\|^2$.*

*Proof.* First, note that $g_r = \nabla J_r(\theta)$ and $g_c = \nabla J_c(\theta)$. Therefore, by assumption and Taylor's theorem, for every $\eta \in [0, 1]$,

$$J_c(\theta + \eta \cdot \Delta) \leq J_c(\theta) + \eta \cdot \langle g_c, \Delta \rangle + \frac{L \cdot \eta^2}{2} \cdot \|\Delta\|^2$$

$$J_r(\theta + \eta \cdot \Delta) \geq J_r(\theta) + \eta \cdot \langle g_r, \Delta \rangle - \frac{L \cdot \eta^2}{2} \cdot \|\Delta\|^2$$

since $\|\eta \cdot \Delta\| \leq M$. By the choice of $\epsilon$ and Lemma 3, $\langle g_c, \Delta \rangle \leq \beta \cdot \langle g_c, \Delta_c \rangle$ and the first claim follows. The second also follows directly by definition of $\Delta$ as a convex combination. Finally, if $g_c = 0$ then $\mu = 0$ and the third claim follows from the second. □

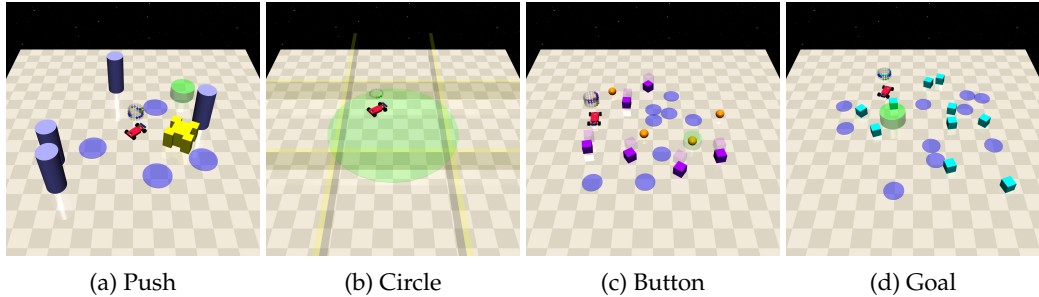

| (a) Push | (b) Circle | (c) Button | (d) Goal |

Figure 3: Safe navigation tasks of Safety Gymnasium (Ji et al., 2023) (images taken from `https://safety-gymnasium.readthedocs.io/en/latest/environments/safe_navigation.html`)

Table 3: Overview of the Safety Gymnasium navigation tasks used in our experiments. The dimensionality refers to the state space for the Point and Car agents, respectively. (Recall that we always use level 2 of the tasks, which is the most difficult.)

| Task | Objective | dim. |
|---|---|---|
| Push | Push the box (yellow) to the goal location (green) while avoiding hazards and pillars (both blue). | 76/88 |
| Circle | Circulate around the center of the circular area (green) while remaining within the boundaries (yellow), with rewards highest along the outermost circumference and increasing with speed. | 28/40 |
| Button | Navigate to the correct goal button (orange with green halo) location and press it while avoiding gremlins (purple) and hazards (blue). | 76/88 |
| Goal | Navigate to the goal (green) while avoiding hazards (blue) and vases (teal). | 60/72 |

**Theorem 2** (Performance Improvement). *There exists $\eta \in (0, 1]$ such that*

1. *if $g_c \neq 0$ then $J_c(\theta + \eta \cdot \Delta) < J_c(\theta)$*
2. *if $g_r \neq 0$ and $\langle g_r, \Delta_c \rangle \geq 0$ (in particular, $g_c = 0$) then $J_r(\theta + \eta \cdot \Delta) \geq J_r(\theta)$.*

*Proof.* Since $J_r$ and $J_c$ are smooth, they are $L$-smooth on the bounded set $B_M(\theta)$ for $M := \max\{\|\Delta_r\|, \|\Delta_c\|\}$ and sufficiently large $L > 0$.

If $g_c \neq 0$ then $\langle g_c, \Delta_c \rangle < 0$ and the claim follows from Lemma 5 for sufficiently small $\eta \in (0, 1]$.

Next, suppose that $g_r \neq 0$ and $\langle g_r, \Delta_c \rangle \geq 0$. By definition of $\Delta_r$ in Eq. (3), $\langle g_r, \Delta_r \rangle > 0$. Besides, by Lemma 5,

$$J_r(\theta + \eta \cdot \Delta) \geq J_r(\theta) + \eta \cdot \mu \cdot \langle g_r, \Delta_r \rangle - \frac{L \cdot \eta^2}{2} \cdot \|\Delta\|^2$$

and the claim follows for sufficiently small $\eta \in (0, 1]$. □

## C SUPPLEMENTARY MATERIALS FOR SECTION 5

### C.1 SUPPLEMENTARY MATERIALS FOR SECTION 5.1

#### C.1.1 SAFETY GYMNASIUM TASKS

We provide additional details on the Safety Gymnasium tasks used in our experiments (see (Ji et al., 2023) and `https://safety-gymnasium.readthedocs.io/` for the full details).

Safe Navigation Tasks are described in Table 3 and visualised in Fig. 3. Visual depictions and complete descriptions of the Safe Velocity tasks are available at `https://safety-gymnasium.readthedocs.io/en/latest/environments/safe_velocity.html`.

Table 4: Hyperparameters for SB-TRPO (Ours), PPO-Lag, TRPO-Lag, CPO, and CUP.

| Hyperparameter | SB-TRPO (Ours) With Critic | No Critic | PPO-Lag | TRPO-Lag | CPO | CUP |
|---|---|---|---|---|---|---|
| Discount Factor $\gamma$ | 0.99 | 0.99 | 0.99 | 0.99 | 0.99 | 0.99 |
| Target KL | 0.01 | 0.01 | 0.02 | 0.01 | 0.01 | 0.02 |
| GAE $\lambda$ | 0.95 | – | 0.95 | 0.95 | 0.95 | 0.95 |
| Timesteps per Epoch | 20000 | 20000 | 20000 | 20000 | 20000 | 20000 |
| Policy Hidden Layers | {64, 64} | {64, 64} | {64, 64} | {64, 64} | {64, 64} | {64, 64} |
| Policy Batch Size | 20000 | 20000 | 64 | 20000 | 64 | 20000 |
| Actor Learning Rate | – | – | 0.0003 | – | – | 0.0003 |
| Actor Optimizer | – | – | ADAM | – | – | ADAM |
| Actor Update Iterations | – | – | 40 | – | – | 40 |
| Critic Hidden Layers | {64, 64} | {64, 64} | {64, 64} | {64, 64} | {64, 64} | {64, 64} |
| Critic Batch Size | 128 | – | 64 | 128 | 128 | 64 |
| Critic Learning Rate | 0.001 | – | 0.0003 | 0.001 | 0.001 | 0.0003 |
| Critic Optimizer | ADAM | – | ADAM | ADAM | ADAM | ADAM |
| Critic Update Iterations | 10 | – | 40 | 10 | 10 | 40 |
| Cost Limit | – | – | 0.00 | 0.00 | 0.00 | 0.00 |
| Clip Coefficient | – | – | 0.20 | – | – | 0.20 |
| Conjugate Gradient (C.G.) Iterations | 50 | 50 | – | 15 | 15 | – |
| C.G. Tikhonov Regularisation Coefficient | 0.02 | 0.02 | – | 0.1 | 0.1 | – |
| Update Scaling Steps | 100 | 100 | – | 15 | 15 | – |
| Step Fraction | 0.80 | 0.80 | – | 0.80 | 0.80 | – |
| Lagrangian Initial Value | – | – | 0.001 | 0.001 | – | 0.001 |
| Lagrangian Learning Rate | – | – | 0.035 | 0.035 | – | 0.035 |
| Lagrangian Optimizer | – | – | ADAM | ADAM | – | ADAM |
| CUP $\lambda$ | – | – | – | – | – | 0.95 |
| CUP $\nu$ | – | – | – | – | – | 2.00 |
| SB-TRPO $\beta$ | [0.6, 0.9] | [0.6, 0.9] | – | – | – | – |

Table 5: Choice of safety bias $\beta$ in each task

| | Point Push | Point Button | Point Goal | Car Push | Car Circle | Car Goal | Hopper Velocity | Swimmer Velocity |
|---|---|---|---|---|---|---|---|---|
| $\beta$ | 0.67 | 0.75 | 0.80 | 0.70 | 0.70 | 0.80 | 0.70 | 0.75 |

**Notes on Agents.**

- **Point:** A 2D robot with two actuators for rotation and forward/backward movement.
- **Car:** A 3D robot with two independently driven wheels and a free rear wheel, requiring coordinated steering and movement.
- The Safe Velocity tasks are based on agents from MuJoCo (Todorov et al., 2012) locomotion tasks, which are extensively documented at `https://gymnasium.farama.org/environments/mujoco/`.

### C.1.2 SELECTION OF HYPERPARAMETERS

Table 4 lists all hyperparameters. For the baselines, we use the official SafePO defaults (Ji et al., 2023), setting the cost limit to 0 to target almost-sure safety. For SB-TRPO, most hyperparameters are inherited from related methods (TRPO-Lagrangian and CPO), and critic-related settings are kept identical in critic-inclusive runs. In runs without critics we use Monte Carlo returns, i.e. GAE with $\lambda = 1$. The only novel parameter is the safety bias $\beta$, for which no established default exists. We therefore performed a light grid search over [0.6, 0.9] for each benchmark task. As shown in Fig. 2c in Section 5.3 and Fig. 6 in Section C.3.1, different choices of $\beta$ trace out an approximate linear reward-cost Pareto frontier, while baselines typically underperform relative to this frontier. For the main results, we select a representative $\beta$ value per task to avoid overly conservative policies while remaining nearly feasible (see Table 5). Notably, $\beta \approx 0.7$ serves as a good starting point across all tasks, highlighting the robustness of our approach (see Tables 6, 7).

Table 6: Rewards and Costs, as well as safety probability and safe reward across 8 benchmark tasks (all at level 2). Red highlights no meaningful task completion (negative (safe) reward) and grey data pairs are not pareto-optimal. Furthermore, we bold the individual best results in each category. $\uparrow$ indicates higher values are better and vice versa.

| Method | Point Push Rewards↑ | Point Push Costs↓ | Point Button Rewards↑ | Point Button Costs↓ | Point Goal Rewards↑ | Point Goal Costs↓ | Car Push Rewards↑ | Car Push Costs↓ | Car Circle Rewards↑ | Car Circle Costs↓ | Car Goal Rewards↑ | Car Goal Costs↓ | Hopper Velocity Rewards↑ | Hopper Velocity Costs↓ | Swimmer Velocity Rewards↑ | Swimmer Velocity Costs↓ |
|---|---|---|---|---|---|---|---|---|---|---|---|---|---|---|---|---|
| Ours ($\beta$ in Table 5) | 0.53 ± 0.19 | 9.2 ± 5.9 | 1.9 ± 0.7 | 15 ± 5 | 1.5 ± 0.6 | 16 ± 7 | 0.55 ± 0.16 | 16 ± 8 | 7.8 ± 2.1 | 0.72 ± 2.80 | 1.1 ± 0.5 | 16 ± 6 | 900 ± 248 | 0.0063 ± 0.0127 | 49 ± 27 | 0.87 ± 3.24 |
| Ours ($\beta$ = 0.70) | 0.65 ± 0.18 | 12 ± 7 | 3.2 ± 0.5 | 17 ± 4 | 3.5 ± 0.7 | 18 ± 8 | 0.55 ± 0.16 | 16 ± 8 | 7.8 ± 2.1 | 0.72 ± 2.80 | 2.4 ± 0.7 | 19 ± 7 | 900 ± 248 | 0.0063 ± 0.0127 | 29 ± 12 | 0.56 ± 1.09 |
| Ours ($\beta$ = 0.75) | 0.43 ± 0.21 | 8.0 ± 6.5 | 1.9 ± 0.7 | 15 ± 5 | 2.4 ± 0.6 | 18 ± 6 | 0.53 ± 0.16 | 14 ± 7 | 7.5 ± 1.5 | 0.28 ± 0.66 | 2.1 ± 0.6 | 18 ± 6 | 860 ± 377 | 0.00028 ± 0.00208 | 49 ± 27 | 0.87 ± 3.24 |
| PPO-Lag | −0.47 ± 1.31 | 27 ± 21 | 3.2 ± 1.2 | 76 ± 27 | 2.8 ± 2.2 | 42 ± 18 | −0.20 ± 0.52 | 34 ± 20 | 8.7 ± 2.0 | 22 ± 37 | 2.6 ± 2.5 | 54 ± 29 | 890 ± 507 | 2.9 ± 2.8 | 40 ± 60 | 6.5 ± 7.2 |
| TRPO-Lag | −0.65 ± 0.55 | 9.3 ± 14.5 | −0.72 ± 0.79 | 15 ± 6 | 0.11 ± 0.35 | 17 ± 9 | −0.21 ± 0.27 | 6.3 ± 4.6 | 13 ± 1 | 18 ± 33 | 0.29 ± 0.50 | 16 ± 7 | 160 ± 77 | 2.1 ± 2.6 | 21 ± 16 | 6.3 ± 6.0 |
| CPO | −1.6 ± 1.1 | 5.8 ± 9.6 | −3.2 ± 1.3 | 58 ± 2.8 | −1.5 ± 0.5 | 9.3 ± 8.0 | −1.4 ± 0.3 | 4.2 ± 4.4 | 0.96 ± 0.75 | 0.98 ± 6.63 | −0.89 ± 0.27 | 13 ± 8 | 880 ± 185 | 0.0 ± 0.0 | −3.8 ± 9.4 | 0.016 ± 0.037 |
| CUP | 0.18 ± 0.26 | 25 ± 22 | 5.2 ± 1.1 | 120 ± 19 | 7.2 ± 2.8 | 110 ± 28 | −0.026 ± 0.362 | 85 ± 50 | 9.6 ± 2.5 | 48 ± 86 | 6.1 ± 2.8 | 110 ± 31 | 1400 ± 114 | 0.43 ± 0.35 | 38 ± 58 | 0.14 ± 0.18 |
| CPPO-PID | −3.4 ± 3.3 | 13 ± 17 | −1.8 ± 0.8 | 24 ± 9 | −2.3 ± 0.9 | 17 ± 10 | −0.99 ± 0.78 | 34 ± 28 | 1.3 ± 1.8 | 27 ± 55 | −0.79 ± 0.69 | 23 ± 10 | 1000 ± 228 | 0.081 ± 0.107 | 0.71 ± 10.89 | 0.044 ± 0.081 |
| FOCOPS | −0.29 ± 0.76 | 35 ± 61 | 9.9 ± 2.0 | 88 ± 14 | 6.5 ± 3.3 | 88 ± 37 | 0.069 ± 0.338 | 35 ± 24 | 8.1 ± 1.9 | 3.9 ± 18.2 | 6.0 ± 5.3 | 86 ± 49 | 970 ± 309 | 0.093 ± 0.135 | 20 ± 12 | 3.0 ± 5.7 |
| RCPO | −0.57 ± 0.51 | 6.3 ± 8.8 | −0.72 ± 0.79 | 15 ± 6 | −0.00026 ± 0.41018 | 12 ± 7 | −0.60 ± 0.49 | 19 ± 20 | 11 ± 2 | 13 ± 35 | 0.29 ± 0.50 | 16 ± 7 | 470 ± 361 | 0.97 ± 1.93 | 19 ± 18 | 5.1 ± 6.7 |
| PCPO | −1.4 ± 0.8 | 14 ± 23 | −2.2 ± 0.8 | 8.8 ± 4.7 | −1.6 ± 0.7 | 18 ± 13 | −0.58 ± 0.46 | 17 ± 18 | 12 ± 1 | 4.4 ± 8.5 | −0.95 ± 0.30 | 17 ± 9 | 750 ± 589 | 3.9 ± 4.9 | −18 ± 2 | 70 ± 16 |
| P3O | 0.3 ± 0.1 | 15 ± 11 | −0.0073 ± 0.2670 | 23 ± 8 | 0.05 ± 0.77 | 36 ± 18 | 0.38 ± 0.15 | 28 ± 12 | 7.6 ± 2.1 | 11 ± 17 | 0.73 ± 0.50 | 40 ± 17 | 1700 ± 34 | 8.9 ± 11.7 | 110 ± 58 | 0.22 ± 0.16 |
| C-TRPO | −1.7 ± 1.4 | 7.8 ± 12.2 | −2.3 ± 0.2 | 6.0 ± 3.0 | −2.5 ± 1.3 | 8.7 ± 5.9 | −1.3 ± 0.5 | 8.9 ± 11.5 | 0.56 ± 0.48 | 3.9 ± 7.4 | −2.1 ± 0.6 | 8.5 ± 6.7 | 6.9 ± 3.5 | 0.0 ± 0.0 | −18 ± 8 | 0.058 ± 0.180 |
| C3PO | −0.18 ± 0.21 | 6.3 ± 5.4 | −0.39 ± 0.16 | 7.8 ± 3.5 | −0.056 ± 0.720 | 7.3 ± 5.4 | 0.078 ± 0.160 | 12 ± 6 | 5.1 ± 1.7 | 0.53 ± 1.00 | −0.28 ± 0.44 | 12 ± 10 | 1600 ± 45 | 0.48 ± 0.41 | 85 ± 60 | 0.10 ± 0.13 |

| Method | Point Push Safe Rew.↑ | Point Push Safe Prob.↑ | Point Button Safe Rew.↑ | Point Button Safe Prob.↑ | Point Goal Safe Rew.↑ | Point Goal Safe Prob.↑ | Car Push Safe Rew.↑ | Car Push Safe Prob.↑ | Car Circle Safe Rew.↑ | Car Circle Safe Prob.↑ | Car Goal Safe Rew.↑ | Car Goal Safe Prob.↑ | Hopper Velocity Safe Rew.↑ | Hopper Velocity Safe Prob.↑ | Swimmer Velocity Safe Rew.↑ | Swimmer Velocity Safe Prob.↑ |
|---|---|---|---|---|---|---|---|---|---|---|---|---|---|---|---|---|
| Ours ($\beta$ in Table 5) | 0.53 ± 0.20 | 0.73 ± 0.07 | 1.9 ± 0.8 | 0.47 ± 0.10 | 1.5 ± 0.6 | 0.72 ± 0.08 | 0.55 ± 0.17 | 0.71 ± 0.08 | 7.8 ± 2.1 | 0.99 ± 0.04 | 1.1 ± 0.5 | 0.99 ± 0.01 | 900 ± 247 | 0.99 ± 0.01 | 49 ± 27 | 0.98 ± 0.02 |
| Ours ($\beta$ = 0.70) | 0.66 ± 0.19 | 0.7 ± 0.1 | 3.2 ± 0.6 | 0.41 ± 0.09 | 3.5 ± 0.7 | 0.66 ± 0.10 | 0.55 ± 0.17 | 0.71 ± 0.08 | 7.8 ± 2.1 | 0.99 ± 0.04 | 2.5 ± 0.8 | 0.65 ± 0.09 | 900 ± 247 | 0.99 ± 0.01 | 29 ± 12 | 0.85 ± 0.27 |
| Ours ($\beta$ = 0.75) | 0.42 ± 0.22 | 0.79 ± 0.08 | 1.9 ± 0.8 | 0.47 ± 0.10 | 2.3 ± 0.7 | 0.68 ± 0.07 | 0.54 ± 0.17 | 0.71 ± 0.07 | 7.5 ± 1.5 | 0.99 ± 0.01 | 2.1 ± 0.6 | 0.66 ± 0.08 | 860 ± 378 | 1.0 ± 0.0 | 49 ± 27 | 0.98 ± 0.02 |
| PPO-Lag | −0.49 ± 1.34 | 0.59 ± 0.22 | 3.2 ± 1.3 | 0.18 ± 0.11 | 2.8 ± 2.2 | 0.47 ± 0.13 | −0.20 ± 0.53 | 0.65 ± 0.10 | 8.7 ± 2.2 | 0.86 ± 0.17 | 2.6 ± 2.5 | 0.45 ± 0.16 | 870 ± 542 | 0.58 ± 0.44 | 32 ± 62 | 0.57 ± 0.42 |
| TRPO-Lag | −0.63 ± 0.57 | 0.78 ± 0.16 | −0.71 ± 0.79 | 0.62 ± 0.10 | 0.11 ± 0.36 | 0.79 ± 0.06 | −0.21 ± 0.28 | 0.89 ± 0.04 | 13 ± 1 | 0.81 ± 0.26 | 0.31 ± 0.51 | 0.75 ± 0.07 | 120 ± 115 | 0.6 ± 0.5 | 4.0 ± 14.5 | 0.33 ± 0.42 |
| CPO | −1.6 ± 1.1 | 0.77 ± 0.19 | −3.2 ± 1.2 | 0.76 ± 0.08 | −1.5 ± 0.5 | 0.86 ± 0.05 | −1.4 ± 0.3 | 0.93 ± 0.05 | 0.96 ± 0.76 | 0.99 ± 0.03 | −0.89 ± 0.27 | 0.82 ± 0.09 | 880 ± 182 | 1.0 ± 0.0 | −3.9 ± 9.4 | 0.99 ± 0.03 |
| CUP | 0.18 ± 0.27 | 0.67 ± 0.19 | 2.4 ± 2.3 | 0.033 ± 0.043 | 6.5 ± 2.9 | 0.15 ± 0.09 | −0.013 ± 0.370 | 0.47 ± 0.19 | 9.4 ± 3.0 | 0.75 ± 0.08 | 6.1 ± 2.9 | 0.15 ± 0.10 | 1400 ± 114 | 0.72 ± 0.19 | 38 ± 58 | 0.90 ± 0.10 |
| CPPO-PID | −3.4 ± 3.4 | 0.77 ± 0.20 | −1.3 ± 0.6 | 0.61 ± 0.10 | −1.7 ± 0.3 | 0.78 ± 0.09 | −0.98 ± 0.79 | 0.69 ± 0.15 | 1.5 ± 1.7 | 0.89 ± 0.16 | −0.72 ± 0.58 | 0.73 ± 0.10 | 1000 ± 253 | 0.81 ± 0.21 | −1.5 ± 11.4 | 0.88 ± 0.09 |
| FOCOPS | −0.28 ± 0.78 | 0.67 ± 0.22 | 5.8 ± 3.8 | 0.11 ± 0.05 | 6.8 ± 2.9 | 0.17 ± 0.17 | 0.069 ± 0.351 | 0.67 ± 0.13 | 8.5 ± 1.8 | 0.92 ± 0.14 | 6.1 ± 5.0 | 0.071 ± 0.0257 | 960 ± 334 | 0.82 ± 0.22 | 5.2 ± 14.8 | 0.52 ± 0.19 |
| RCPO | −0.57 ± 0.54 | 0.75 ± 0.17 | −0.96 ± 0.94 | 0.64 ± 0.09 | −0.27 ± 0.78 | 0.80 ± 0.07 | −0.60 ± 0.49 | 0.78 ± 0.11 | 11 ± 2 | 0.86 ± 0.21 | 0.29 ± 0.51 | 0.75 ± 0.08 | 450 ± 388 | 0.74 ± 0.28 | 8.4 ± 20.0 | 0.61 ± 0.18 |
| PCPO | −1.4 ± 0.8 | 0.78 ± 0.19 | −1.9 ± 0.6 | 0.70 ± 0.10 | −1.2 ± 0.3 | 0.78 ± 0.09 | −0.58 ± 0.46 | 0.8 ± 0.1 | 12 ± 2 | 0.89 ± 0.14 | −0.91 ± 0.27 | 0.78 ± 0.09 | 720 ± 615 | 0.71 ± 0.31 | −15 ± −4 | 0.80 ± 0.00 |
| P3O | 0.3 ± 0.2 | 0.71 ± 0.14 | −0.19 ± 0.58 | 0.60 ± 0.09 | 0.09 ± 0.09 | 0.61 ± 0.10 | 0.38 ± 0.17 | 0.72 ± 0.08 | 7.3 ± 2.6 | 0.91 ± 0.10 | 0.70 ± 0.56 | 0.59 ± 0.11 | 1700 ± 41 | 0.43 ± 0.46 | 110 ± 58 | 0.91 ± 0.09 |
| C-TRPO | −1.7 ± 1.4 | 0.78 ± 0.15 | −1.7 ± 0.1 | 0.72 ± 0.08 | −1.6 ± 0.3 | 0.84 ± 0.07 | −1.3 ± 0.5 | 0.86 ± 0.07 | 0.69 ± 0.45 | 0.95 ± 0.07 | −2.0 ± 0.4 | 0.83 ± 0.08 | −40 ± 24 | 0.93 ± 0.05 | −18 ± 8 | 0.91 ± 0.09 |
| C3PO | −0.17 ± 0.21 | 0.85 ± 0.05 | −0.39 ± 0.18 | 0.76 ± 0.07 | −0.046 ± 0.735 | 0.86 ± 0.06 | 0.077 ± 0.168 | 0.86 ± 0.05 | 5.1 ± 1.7 | 0.99 ± 0.01 | −0.29 ± 0.45 | 0.86 ± 0.06 | 1600 ± 47 | 0.79 ± 0.14 | 85 ± 60 | 0.92 ± 0.08 |

Table 7: Safety-biased–cost–reward (SCR) metric across 8 benchmark tasks.

| Method | Point Push | Point Button | Point Goal | Car Push | Car Circle | Car Goal | Hopper Velocity | Swimmer Velocity |
|---|---|---|---|---|---|---|---|---|
| Ours ($\beta$ in Table 5) | 0.037 | 0.058 | 0.064 | 0.023 | 4.5 | 0.047 | 890 | 26 |
| Ours ($\beta$ = 0.70) | 0.035 | 0.073 | 0.12 | 0.023 | 4.5 | 0.083 | 890 | 16 |
| Ours ($\beta$ = 0.75) | 0.037 | 0.058 | 0.083 | 0.025 | 5.9 | 0.071 | 860 | 26 |
| PPO-Lag | -0.010 | 0.0075 | 0.031 | -0.0037 | 0.33 | 0.021 | 130 | 2.4 |
| TRPO-Lag | -0.048 | -0.027 | 0.0051 | -0.025 | 0.57 | 0.013 | 23 | 0.18 |
| CPO | -0.18 | -0.36 | -0.12 | -0.25 | 0.48 | -0.051 | 880 | -3.7 |
| CUP | 0.0047 | 0.00064 | 0.0085 | -0.000071 | 0.14 | 0.0087 | 720 | 30 |
| CPPO-PID | -0.19 | -0.032 | -0.077 | -0.019 | 0.048 | -0.022 | 770 | -1.3 |
| FOCOPS | -0.0053 | 0.0069 | 0.013 | 0.0013 | 1.6 | 0.005 | 720 | 0.67 |
| RCPO | -0.058 | -0.038 | -0.017 | -0.023 | 0.71 | 0.013 | 170 | 0.84 |
| PCPO | -0.075 | -0.14 | -0.049 | -0.026 | 2.0 | -0.039 | 100 | -0.19 |
| P3O | 0.013 | -0.0048 | 0.0015 | 0.0093 | 0.53 | 0.010 | 75 | 81 |
| C-TRPO | -0.15 | -0.18 | -0.14 | -0.11 | 0.13 | -0.17 | -37 | -15 |
| C3PO | -0.020 | -0.034 | -0.0048 | 0.005 | 3.3 | -0.02 | 870 | 71 |

C.1.3 Reproduction of Results

All experiments were run with `Python 3.8`. To reproduce our results, first install Safety-Gymnasium as documented at `https://github.com/PKU-Alignment/safety-gymnasium#installation`, and then extract the supplementary code provided with this submission. Inside the extracted folder, install SafePO (Ji et al., 2023) integrated with our approach by running:

```
pip install -e .
```

Experiments can be reproduced in two ways. To train a single baseline (e.g., PPO-Lagrangian) with a specific seed (e.g., 2000) on a given environment (e.g., SafetyPointGoal2-v0), navigate to the `single_agent` directory and run:

```
python ppo_lag.py --task SafetyPointGoal2-v0 --seed 2000 --cost_limit 0
```

Similarly, for our method, SB-TRPO, at a specific safety bias (e.g., $\beta = 0.65$), use:

```
python sb-trpo.py --task SafetyPointGoal2-v0 --seed 2000 --beta 0.65
```

Additional options can be found in `utils/config.py` under `single_agent_args`.

For large-scale benchmarking, multiple algorithms can be launched in parallel with a single command:

```
python benchmark.py
--tasks SafetyHopperVelocity-v1 SafetySwimmerVelocity-v1
--algo ppo_lag trpo_lag cpo cup sb-trpo
```

```
--workers 25 --num-envs 20 --steps-per-epoch 20000
--total-steps 20000000 --num-seeds 5
--cost-limit 0 --beta 0.7
```

This command runs all baselines considered in the paper at a cost limit of 0, alongside our method with $\beta = 0.7$, using 25 training processes in parallel with 20 vectorized environments per process and 5 random seeds. Further options are documented in `benchmark.py`.

Finally, our ablations are implemented as `sb-trpo_rcritic` (including a reward critic) and `sb-trpo_critic` (including both reward and cost critics).

## C.2 SUPPLEMENTARY MATERIALS FOR SECTION 5.2

The training curves, which are smoothed with averaging over the past 20 epochs, are presented in Fig. 4.

Additionally, we train longer on Point Button to confirm further feasibility and reward improvements. This is presented in Fig. 5. As anticipated, the improvements are non-diminishing even for a 125% longer training run.

We compare the compute costs of our approach with the baselines w.r.t. the update times per epoch on a single, non-parallelized training run in Table 8. This demonstrates that training critics is expensive across all critic-integrated approaches.

### C.2.1 QUALITATIVE BEHAVIOUR AND VIDEOS

We compare the qualitative behaviours of policies obtained from our approach with the baselines using recorded videos (provided in the supplementary material) across four benchmark tasks: Point Button, Car Circle, Car Goal, and Swimmer Velocity. The results in Tables 1 and 2 are strongly corroborated by the behaviours observed in these videos.

**Point Button.** Most approaches exhibit low reward affinity and consequently engage in overconservative behaviours, such as moving away from interactive regions in the environment. Among the baselines, CUP, PPO-Lag, and SB-TRPO show visible reward-seeking behaviours, in decreasing order of intensity. CUP sometimes ignores obstacles entirely and is confused by moving obstacles (Gremlins). PPO-Lag avoids Gremlins more intelligently but is not particularly creative in approaching the target button. In contrast, SB-TRPO exhibits a creative strategy: it slowly circles the interactive region, navigating around moving obstacles to reach the target button.

**Car Circle.** CPO and C-TRPO fail to fully associate circling movement with reward acquisition, resulting in back-and-forth movements near the circle's centre and partial rewards. Consequently, these policies rarely violate constraints. C3PO learns to circle in a very small radius to acquire rewards, but its circling behaviour is intermittent. P3O, CUP, and PPO-Lag learn to circle effectively but struggle to remain within boundaries, incurring costs. TRPO-Lag and SB-TRPO achieve both effective circling and boundary adherence.

**Car Goal.** CPO, C3PO, and C-TRPO display low reward affinity, leading to overconservative behaviours such as avoiding the interactive region. TRPO-Lag is unproductive and does not avoid hazards reliably. CUP and PPO-Lag are highly reward-driven but reckless with hazards. P3O begins to learn safe navigation around the interactive region, though occasional recklessness remains. SB-TRPO achieves productive behaviour while intelligently avoiding hazards, e.g., by slowly steering around obstacles or moving strategically within the interactive region.

**Swimmer Velocity.** CPO and C-TRPO sometimes move backward or remain unproductively in place. TRPO-Lag and PPO-Lag move forward but engage in unsafe behaviours. CUP and SB-TRPO are productive in moving forward while being less unsafe. C3PO and P3O are the most effective, moving forward consistently while exhibiting the fewest unsafe behaviours.

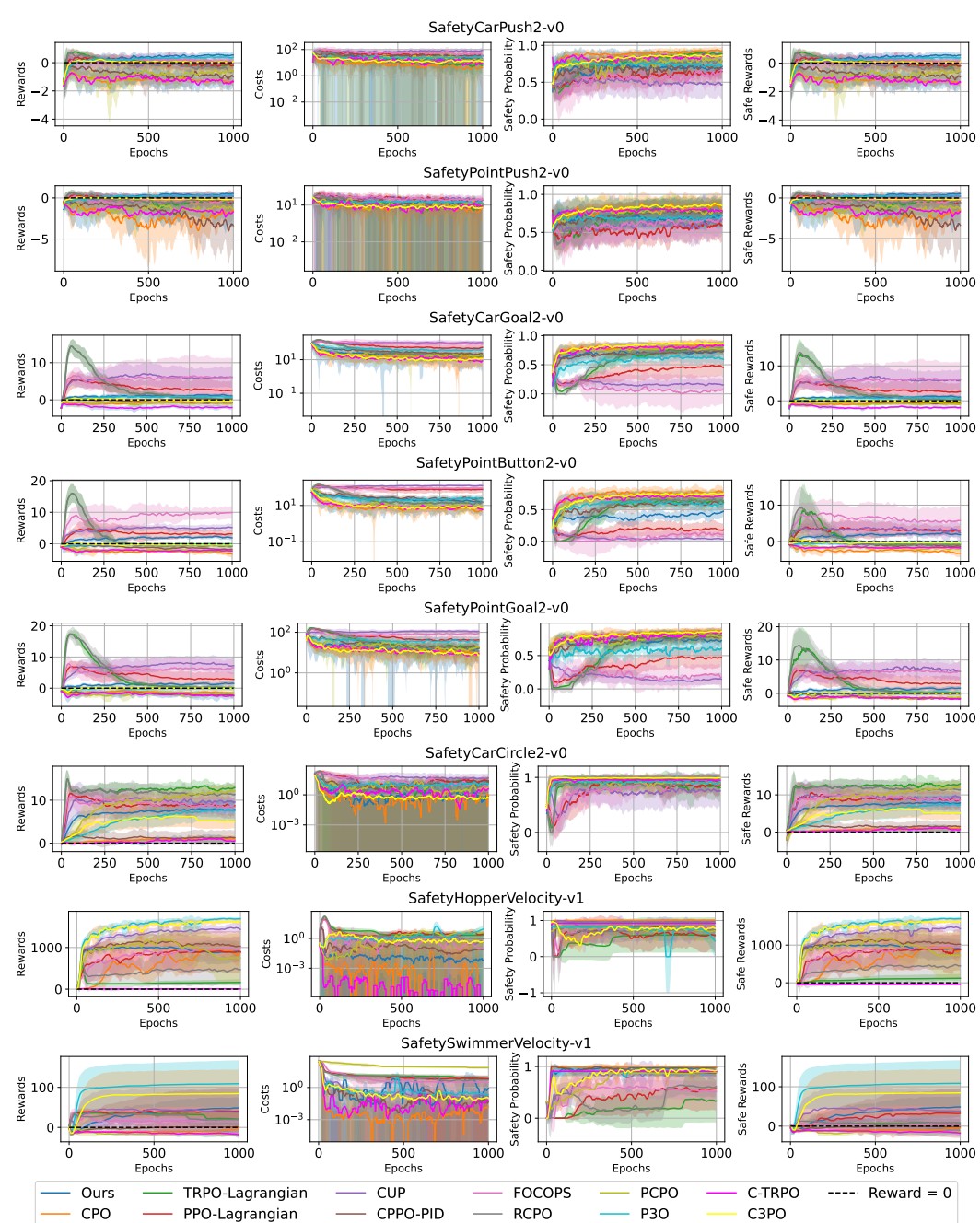

Figure 4: Training curves

C.2.2 Deviations from Theory in Training Curves

While our theoretical results (Theorem 2) guarantee consistent improvement in both reward and cost for sufficiently small steps, practical training exhibits deviations due to finite sample estimates and the sparsity of the cost signal in some tasks. We summarise task-specific observations:

- In **Swimmer Velocity**, some update steps may not immediately reduce incurred costs due to estimation noise, which explains delays in achieving perfect feasibility.

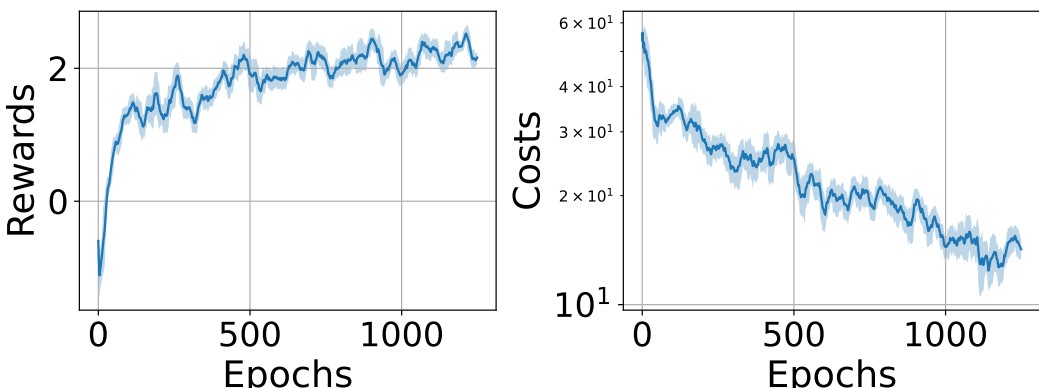

Figure 5: Point Button: training longer, doing better.

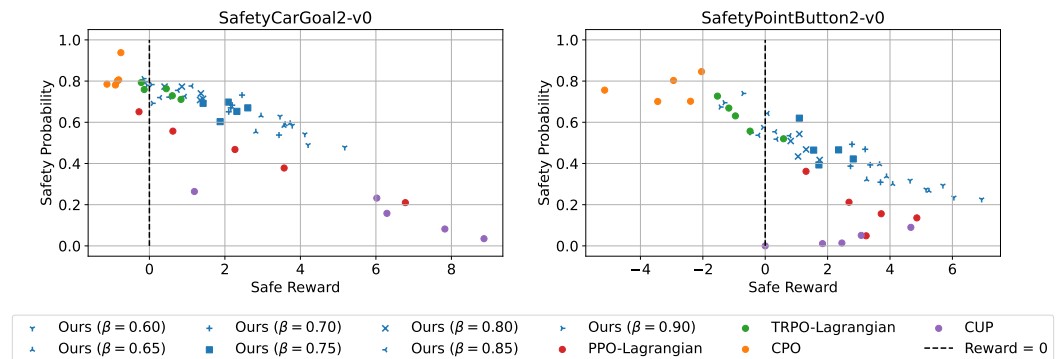

Figure 6: Ablation study of safety bias $\beta$

- In **Car Circle**, policies that previously attained near-zero cost sometimes return temporarily to infeasible behaviour. This arises because the update rule does not explicitly encode "distance" from infeasible regions within the feasible set. Nonetheless, policies eventually return to feasibility thanks to the guarantees of our approach.
- In **Hopper Velocity**, after quickly achieving high reward, performance shows a slow decline accompanied by gradual improvements in cost. This reflects the prioritisation of feasibility improvement over reward during some updates, without drastically compromising either objective.
- In **navigation tasks**, especially **Button, Goal, and Push**, the cost signal is sparse: a cost of 1 occurs only if a hazard is touched. This sparsity makes perfect safety more difficult to achieve, as most updates provide limited feedback about near-misses or almost-unsafe trajectories. Despite this, SB-TRPO steadily reduces unsafe visits and substantially outperforms baselines.

Overall, these deviations are a natural consequence of finite sample estimates, quadratic approximations, and sparse cost signals. In spite of this, SB-TRPO consistently improves reward and safety together, achieving higher returns and lower violations than all considered baselines.

C.3 Supplementary Materials for Section 5.3

C.3.1 Effect of Safety Bias $\beta$

We present additional scatter plots for safety probability versus safe reward for our approach compared to the baselines in Fig. 6. We also compare our approach at specific $\beta$

Table 8: Update times per epoch on Point Goal on a single, non-parallelized training run, based on the hyperparameters in Table 4. Mean ± STDEV is reported, rounded to 2 decimal places. The least update time consumed per epoch is in **bold**. Note that PPO-Lagrangian, in contrast to TRPO-Lagrangian, has a budget of up to 40 gradients steps per epoch (as per Table 4).

| | SB-TRPO (No Critic) | SB-TRPO (With Reward Critic) | SB-TRPO (With Both Critics) | TRPO-Lag | CPO | PPO-Lag | CUP |
|---|---|---|---|---|---|---|---|
| Time$^{\downarrow}$ | **0.26 ± 0.08** | 5.76 ± 0.91 | 8.12 ± 0.95 | 5.85 ± 0.79 | 6.59 ± 1.96 | 69.83 ± 8.29 | 84.60 ± 15.54 |

Table 9: Mean metrics across Point Button and Car Goal to study the effect of critic on SB-TRPO. Values are rounded to 2 significant digits. Best performer per task is in **bold**.

| | Point Button ($\beta = 0.75$) | | | Car Goal ($\beta = 0.70$) | | |
|---|---|---|---|---|---|---|
| | both critics | reward critic | without critics | both critics | reward critic | without critics |
| Rewards$^{\uparrow}$ | **5.0** | 2.5 | 1.9 | **5.9** | 5.3 | 2.5 |
| Costs$^{\downarrow}$ | 36 | 18 | **15** | 38 | 30 | **19** |
| Safety Probability$^{\uparrow}$ | 0.29 | 0.39 | **0.47** | 0.40 | 0.39 | **0.65** |
| Safe Rewards$^{\uparrow}$ | **5.0** | 2.4 | 1.9 | **5.9** | 5.3 | 2.5 |
| SCR$^{\uparrow}$ | 0.038 | 0.051 | **0.058** | 0.061 | 0.067 | **0.083** |

values across all benchmark tasks with the baselines (Tables 6 and 7). SB-TRPO consistently achieves the best balance of performance and feasibility across all benchmark tasks, demonstrating robustness to the choice of $\beta$.

### C.3.2 Effect of Critic

The values of metrics regarding the ablation study about the effect of critics are provided in Tables 8 and 9.

## D Novelty and Comparison to Existing Methods

We refer the reviewer to our re-written derivation of SB-TRPO as a generalisation of CPO overcoming its greedy behaviour in the cost-focused recovery phase, which is triggered very frequently in the setting of hard constraints. We proceed by highlighting our core contribution and conceptual novelty, and explaining why these lead to superior performance in practice.

**Core contribution and conceptual novelty.** Whilst prior work usually addresses general CMDPs, our method targets the setting with hard constraints specifically, allowing it to excel in it. Although our method builds on the trust-region framework of TRPO, the key conceptual novelty lies in introducing generalised update rule capturing controlled, simultaneous progress on both safety and reward at every iteration. Specifially, the cost reduction per update is

$$\epsilon := -\beta \cdot \langle g_c, \Delta_c \rangle,$$

where $\Delta_c$ is the optimal solution to the cost-only linearised trust-region problem Eq. (2). This design has three important consequences.

- *No separate recovery phase.* Unlike CPO and C-TRPO, our algorithm never switches into a feasibility-restoration mode. Safety and reward are addressed within the *same* update, producing smoother learning dynamics and avoiding the abrupt, purely cost-driven behaviour characteristic of recovery phases.

- *CPO behaviour is recovered for $\beta = 1$.* For $\beta = 1$, the safety requirement matches exactly the zero-threshold constraint imposed by CPO, so our update includes CPO-like behaviour as a special case. For $\beta < 1$, the method is deliberately less aggressive in driving the policy toward feasibility, which—as our experiments show—prevents collapse into overly conservative, trivially safe states.

- *Guaranteed improvement of both cost and reward.* As formalised in Theorem 2, the update guarantees (i) strict cost improvement whenever $g_c \neq 0$, and (ii) strict reward improvement whenever the reward and cost gradients are not misaligned. Thus, reward improvement is guaranteed even *before* feasibility is achieved—something CPO and related methods such as C-TRPO do not guarantee during its feasibility recovery/attainment phase.

Moreover, compared to Lagrangian methods, which update via $\Delta_r + \lambda \cdot \Delta_c$ with $\lambda$ that *does not take the current reward and cost updates $\Delta_r$ and $\Delta_c$ into account* and increases monotonically under zero-cost thresholds, these methods typically either over- or under-emphasise safety. By contrast, we dynamically weight $\Delta_r$ and $\Delta_c$ with $\mu$ defined in Eq. (4), providing the aforementioned formal improvement guarantees for both reward and cost—*something Lagrangian methods do not offer*.

In summary, our contribution is not a basic penalty modification of TRPO but a principled, guaranteed-improvement update rule for hard-constrained CMDPs that subsumes CPO as a special case while avoiding its limitations, notably its collapse into over-conservatism.

**Why CPO behaves poorly in hard-constraint settings.** CPO attempts to maintain feasibility at every iteration. When constraints are violated, it prioritises cost reduction to restore feasibility. In hard-constraint regimes, this often drives the policy into *trivially safe* regions (e.g., corners without hazards), which are far from the states where reward can be obtained (e.g., far from goals in navigation tasks). Escaping such regions typically requires temporary constraint violations, which CPO forbids, leading to stagnation. Moreover, CPO and related methods such as C-TRPO *do not provide reward-improvement guarantees during this feasibility-recovery phase*.

**Why our method behaves better.** Because SB-TRPO is *less greedy about attaining feasibility*, it avoids collapsing into trivially safe but task-ineffective regions. Besides, by *not switching between separate phases for reward improvement and feasibility recovery* (as in CPO), our method exhibits markedly *smoother learning dynamics*.

Moreover, Fig. 7 shows that SB-TRPO's update directions align well with *both* the cost and reward gradients, whereas CPO's updates are typically at best orthogonal to the reward gradient. By Theorem 2, such alignment *guarantees consistent progress in both safety and task performance*.

In summary, SB-TRPO yields *smoother learning*, avoids collapse into conservative solutions, and enables *steady reward accumulation* over iterations. Empirically, this leads to substantially higher returns than CPO while still maintaining very low safety violations.

### D.1 CONNECTION TO C-TRPO

**Update Directions and Geometries.** C-TRPO (Milosevic et al., 2024) modifies the TRPO trust-region *geometry* by adding a barrier term that diverges near the feasible boundary. This yields a deformed divergence that increasingly penalises steps pointing towards constraint violation. When the constraint is violated, C-TRPO enters a dedicated *recovery phase* in which the update direction becomes the pure cost gradient; reward improvement is not pursued during this phase.

SB-TRPO keeps the standard TRPO geometry fixed and instead modifies the *update direction* itself: the step is the solution of a two-objective trust-region problem with a required fraction $\beta$ of the optimal local cost decrease. This yields a *dynamic convex combination* of the optimal reward and cost directions. Unlike C-TRPO:

- there is **no separate recovery phase**,
- the update direction is always a mixture of $(\Delta_r, \Delta_c)$ rather than switching geometries, and
- *reward improvement remains possible in every step* whenever the reward and cost gradients are sufficiently aligned.

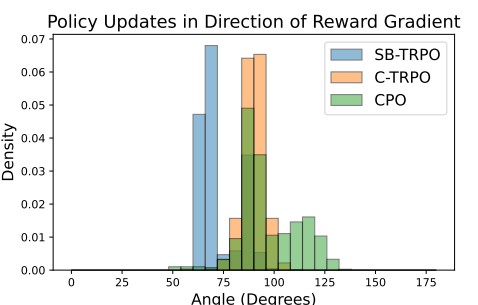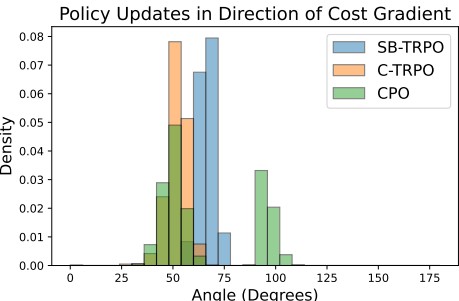

Figure 7: Car Circle: Orientations of Policy Updates to Reward and Cost Gradients

This clarifies the relation between the two approaches: C-TRPO mixes *geometries* (via a KL barrier that reshapes the trust region), while SB-TRPO dynamically mixes *update directions*. In particular, during recovery C-TRPO uses $\Delta_c$, whereas SB-TRPO continues to use $\Delta$, the solution of the trust-region problem considering both reward and cost; see Fig. 1.

Finally, the safety-bias parameter $\beta$ in SB-TRPO plays a role fundamentally different from the barrier coefficient in C-TRPO: $\beta$ determines the minimum fraction of the optimal local cost decrease required from each step, while the barrier coefficient in C-TRPO controls the strength of the geometric repulsion from the boundary.

**Angles between Update Direction and Cost/Reward Gradients.** We plot the angles between policy updates and cost/reward gradients in Figure 7. We observe that for CPO and C-TRPO most angles between update direction and reward gradient are 90° or larger, whilst for SB-TRPO most angles are below 90° with a concentration around 60° − 70°. In the light of Theorem 2, this almost always ensures a reward improvement (for sufficiently small step size).

On the other hand, angles between update direction of SB-TRPO and cost gradient tend to be larger than for CPO and C-TRPO, illustrating its less greedy feasibility attainment.

**Connection to CPO.** Both algorithms can be seen to generalise CPO in different ways:

- **C-TRPO:** Milosevic et al. (2024) show in Prop. 4.2 that the approximate C-TRPO update approaches the CPO update as the coefficient of the barrier term vanishes, recovering CPO in the limit. Besides, both CPO and C-TRPO have an explicit recovery phase when the update is infeasible.
- **SB-TRPO:** As argued in Section 4.1, setting $\beta = 1$ in SB-TRPO recovers CPO exactly and elegantly captures both the recovery and feasible update phases.

**Theoretical Guarantees.** Both algorithms have strong theoretical guarantees:

- **C-TRPO** provides
  - CPO-style "almost" reward-improvement guarantees for steps within the feasible region under the barrier geometry,
  - bounds on constraint violations for feasible steps, and
  - a time-continuous global convergence result under fairly strong assumptions (finite state/action spaces and "regular" policies (Milosevic et al., 2024, Sec. 4.2)).

  Outside the feasible region (during the recovery phase), *reward improvement is not guaranteed*.
- **SB-TRPO** guarantees for sufficiently small step sizes in the quadratic-approximation regime (Theorem 2):
  1. cost strictly decreases at every iteration (unless cost gradient vanishes),
  2. reward strictly improves whenever reward and cost gradients are sufficiently aligned.

In particular, there is no recovery phase: *even if cost is high, updates still improve reward* (as long gradients are not misaligned).

Besides, in the idealised setting of (**Update 1**), if neither cost nor reward improves further, the method arrives at a *trust-region local optimum for cost* and a *trust-region local optimum for the constrained problem* with the modified cost threshold (Theorem 1).

In short, C-TRPO guarantees approximate reward improvement only within feasible steps, whereas SB-TRPO can guarantee reward improvement even for currently infeasible policies.

**Empirical Performance.** We added C-TRPO (as well as C3PO and others) to our baselines and present results in Tables 1 and 2. We can observe that C-TRPO often achieves slightly lower cost than SB-TRPO, which however typically comes at the expense of negative or significantly lower rewards, resulting in poor task performance.

