# OpenReview forum: "Safety-Biased Policy Optimisation: Towards Hard-Constrained Reinforcement Learning via Trust Regions"
_ICLR.cc/2026/Conference — Submitted to ICLR 2026_

### Official Review · Reviewer_XtT1 · 2025-10-26

**Soundness:** 2
**Presentation:** 3
**Contribution:** 2
**Rating:** 2
**Confidence:** 3

**Summary:**

Existing safe RL methods either fail to ensure near-zero safety violations or severely degrade reward performance in the hard-constraint setting. The paper proposes extending TRPO to the CMDP setting. The key idea is not to explicitly enforce that the cost is zero (or below a fixed threshold) at every update. Instead, the method introduces a surrogate cost constraint requiring that the cost of the new policy be no greater than that of the previous policy. The algorithm then uses the conjugate gradient method from TRPO to separately compute the update directions for the reward objective and for the surrogate cost objective. It then forms a convex combination of these gradient directions. The paper proves that this convex combination maximizes a linearized objective that balances reward improvement and surrogate cost reduction.

**Strengths:**

1.The paper extends TRPO to a hard-constraint setting and provides a proof of a performance improvement guarantee.

2.It clearly discusses the difficulty of safe RL under hard constraints in CMDPs, and shows that many existing methods fail in this regime.

3.The proposed method is conceptually simple and appears easy to implement.

**Weaknesses:**

1. CPO already extends TRPO to constrained MDP (CMDP) settings. Therefore, the paper should make explicit what the key differences are between CPO and the proposed method, and which design choices lead to better performance in practice. I did not find a clear discussion of this.

2. My understanding is that CPO provides a performance improvement guarantee, but still performs poorly in hard-constraint settings. This may be due to its approximate solution procedure and the fact that it tries to maintain feasibility at every step. The proposed method also claims a performance improvement guarantee, but I do not see a clear explanation of why it would behave better than CPO in practice under strict (near-zero) safety constraints. This needs to be articulated more clearly.

3. The training curves in Figure 4 suggest that the proposed method still does not consistently achieve both near-zero safety violations and high reward in the hard-constraint setting. It appears that the problem is not fully solved. Please explain, task by task, what is happening in Figure 4 (for example, when cost remains non-zero or reward collapses).

**Questions:**

1. Why do you not report a metric that reflects task completion under safety, such as the cost per successful episode (i.e., episodes that actually reach the goal / complete the task)? The “safety probability” and “safe rewards” metrics you define focus on constraint satisfaction, but if the agent fails to complete the task, then having near-zero cost is not very meaningful. In many safe RL algorithms, when the constraint is enforced as a hard (near-zero) constraint, the agent simply fails to solve the task at all.

2. I am familiar with the Safety Gymnasium tasks. In your Table 1, many tasks have very low or even negative reward values (e.g., Point Push, Point Button, Point Goal, Car Push, Car Circle, Car Goal). This suggests that the policies often fail to complete the task. How should we interpret “safety probability” in such cases? Is the agent just behaving very conservatively and not solving the task?

3. There are no video rollouts or qualitative demonstrations comparing different methods. This makes it hard to judge whether the learned policies are actually completing the tasks versus just being safe by doing nothing. Please provide qualitative evidence (e.g., videos or behavioral descriptions) to show what the learned policies are doing for each algorithm.

4. In Lemma 3, you assume ⟨g_c, Δ_c⟩ ≤ −ϵ. This assumption may not hold in general. Could you justify why this is reasonable in practice? As written, it is not obvious to me that this condition always holds, and the later argument depends critically on it.

---

> ### Author Response · Authors · 2025-11-23
>
> We thank the reviewer for their valuable insights. We refer them to the general response for a clarification of the novelty of our method and a discussion of the design choices that lead to its strong empirical performance. There, we also explain why our performance improvement guarantee, covering *both* reward and cost, is strictly stronger: unlike CPO, which uses recovery steps with no guarantees about reward, our guarantee holds for *every* update.
>
> Below, we address the reviewer’s specific questions concerning the assumption in Lemma 3, the evaluation metrics and the low rewards observed for some baselines, as well as the qualitative behaviour in training curves/videos and the degree of progress achieved.
>
> # Justification of the assumption in Lemma 3
> We hope this has been clarified by the rewritten derivation of our method.
>
> Specifically, in our algorithm $\epsilon$ is chosen dynamically at each update step as $\epsilon := -\beta \cdot \langle g_c, \Delta_c \rangle$, where $\Delta_c$ is an optimal solution to the cost-only objective (Eq. 2) and $\beta \in (0,1]$ is the safety bias. Due to $\langle g_c, \Delta_c \rangle \le 0$ it holds that $\langle g_c, \Delta_c \rangle \le -\epsilon$, i.e. the assumption in Lemma 3 is automatically satisfied for our choice of $\epsilon$.
>
> Intuitively, this ensures that the cost decreases by only a fraction of the purely cost-directed step. This adaptive choice of $\epsilon$ ensures that the assumption in Lemma 3 holds at every update while still allowing the convex combination with the reward-directed direction to improve reward.
>
> # Metrics and Low Rewards by Baselines
> We thank the reviewer for the suggestion. The metric "cost per successful episode" is indeed meaningful in settings where constraint violations are not safety-critical and instead represent, for instance, resource usage or budget consumption. In these domains, ignoring unsafe trajectories and averaging only over successful ones can give a useful view of efficiency conditioned on achieving the task.
>
> However, our work focuses on safety-critical problems in which *any* constraint violation is undesirable irrespective of whether the episode ultimately succeeds. In such settings, conditioning on successful episodes can obscure unsafe behaviour: an agent that repeatedly attempts the task unsafely and succeeds only occasionally would appear favourable under a "cost per completed task" metric, despite being unsafe in the majority of its operation.
>
> For this reason, our evaluation metrics emphasise both overall constraint satisfaction ("safety probability") and *safe task performance*, captured by the "safe reward" which measures the reward accumulated exclusively along fully safe trajectories. This directly reflects task completion *under safety*, so that unsafe episodes do not contribute positively.
>
> An essential property in these domains is to achieve the task *fairly often* while remaining *fairly safe*, something that current baselines often fail to do.
>
> We therefore regard cost per successful episode as complementary but not aligned with the safety-critical perspective taken in this paper, and we leave its study to future work focused on non-safety-critical domains within the standard CMDP setup with positive cost thresholds, where it is more appropriate.

---

> > ### Author Response · Authors · 2025-11-23
> >
> > # Qualitative Behaviour and Videos
> > We compare the qualitative behaviours of policies obtained from our approach with the baselines using recorded videos (provided in the supplementary material) across four benchmark tasks: Point Button, Car Circle, Car Goal, and Swimmer Velocity. The results in Tables 1 and 2 are strongly corroborated by the behaviours observed in these videos.
> >
> > **Point Button.** Most approaches exhibit low reward affinity and consequently engage in overconservative behaviours, such as moving away from interactive regions in the environment. Among the baselines, CUP, PPO-Lag, and SB-TRPO show visible reward-seeking behaviours, in decreasing order of intensity. CUP sometimes ignores obstacles entirely and is confused by moving obstacles (Gremlins). PPO-Lag avoids Gremlins more intelligently but is not particularly creative in approaching the target button. In contrast, SB-TRPO exhibits a creative strategy: it slowly circles the interactive region, navigating around moving obstacles to reach the target button.
> >
> > **Car Circle.** CPO and C-TRPO fail to fully associate circling movement with reward acquisition, resulting in back-and-forth movements near the circle’s centre and partial rewards. Consequently, these policies rarely violate constraints. C3PO learns to circle in a very small radius to acquire rewards, but its circling behaviour is intermittent. P3O, CUP, and PPO-Lag learn to circle effectively but struggle to remain within boundaries, incurring costs. TRPO-Lag and SB-TRPO achieve both effective circling and boundary adherence.
> >
> > **Car Goal.** CPO, C3PO, and C-TRPO display low reward affinity, leading to overconservative behaviours such as avoiding the interactive region. TRPO-Lag is unproductive and does not avoid hazards reliably. CUP and PPO-Lag are highly reward-driven but reckless with hazards. P3O begins to learn safe navigation around the interactive region, though occasional recklessness remains. SB-TRPO achieves productive behaviour while intelligently avoiding hazards, e.g., by slowly steering around obstacles or moving strategically within the interactive region.
> >
> > **Swimmer Velocity.** CPO and C-TRPO sometimes move backward or remain unproductively in place. TRPO-Lag and PPO-Lag move forward but engage in unsafe behaviours. CUP and SB-TRPO are productive in moving forward while being less unsafe. C3PO and P3O are the most effective, moving forward consistently while exhibiting the fewest unsafe behaviours.
> >
> >
> > # Deviations from Theory in Training Curves
> > While our theoretical results (Theorem 2) guarantee consistent improvement in both reward and cost for sufficiently small steps, practical training exhibits deviations due to finite sample estimates and the sparsity of the cost signal in some tasks. We summarise task-specific observations:
> >
> > * In **Swimmer Velocity**, some update steps may not immediately reduce incurred costs due to estimation noise, which explains delays in achieving perfect feasibility.
> > * In **Car Circle**, policies that previously attained near-zero cost sometimes return temporarily to infeasible behaviour. This arises because the update rule does not explicitly encode "distance" from infeasible regions within the feasible set. Nonetheless, policies eventually return to feasibility thanks to the guarantees of our approach.
> >  * In **Hopper Velocity**, after quickly achieving high reward, performance shows a slow decline accompanied by gradual improvements in cost. This reflects the prioritisation of feasibility improvement over reward during some updates, without drastically compromising either objective.
> >   * In **navigation tasks**, especially **Button, Goal, and Push**, the cost signal is sparse: a cost of 1 occurs only if a hazard is touched. This sparsity makes perfect safety more difficult to achieve, as most updates provide limited feedback about near-misses or almost-unsafe trajectories. Despite this, SB-TRPO steadily reduces unsafe visits and substantially outperforms baselines.
> >
> > Overall, these deviations are a natural consequence of finite sample estimates, quadratic approximations, and sparse cost signals. In spite of this, SB-TRPO consistently improves reward and safety together, achieving higher returns and lower violations than all considered baselines.

---

> > > ### Author Response · Authors · 2025-11-23
> > >
> > > # Clarifying the Degree of Progress Achieved
> > > We agree with the reviewer that our method does not yet *fully* solve the hard-constraint problem in the sense of simultaneously achieving perfectly zero violations and near-optimal reward on every task. This is extremely challenging in environments such as navigation problems in Safety Gymnasium at level 2. Despite significant research effort, no existing model-free safe RL method achieves this consistently, partially because the focus has been on methods for CMDPs with *positive cost thresholds*.
> > >
> > > However, our results demonstrate *substantial and consistent progress* over prior approaches.
> > > Thus, while the hard-constraint problem is not completely solved, our method significantly advances the state of the art: it advocates zero-cost problems and is the first approach that robustly reduces safety violations even on the harder environments without sacrificing reward to the same extent as existing baselines.

---

### Official Review · Reviewer_1Wi6 · 2025-10-27

**Soundness:** 2
**Presentation:** 3
**Contribution:** 2
**Rating:** 2
**Confidence:** 4

**Summary:**

The manuscript proposes a new optimizer for constrained Markov decision processes, which constitute an important model for safe reinforcement learning. The new method, called Safety-Biased Trust Region Policy Optimization (SB-TRPO) updates the policy's parameters according to an adaptive convex combination of the natural policy gradients of the reward and cost. Both, a theoretical analysis and empirical comparison to some existing methods are presented.

**Strengths:**

+ Very well writte, easy to understand and parse.
+ Addresses a timely and important problem of safe policy optimization.
+ Proposes a new way of adaptively weighting reward and cost natural policy gradients.
+ Clean and simple method, which provides promising empirical results.

**Weaknesses:**

I mainly see one weakness: The manuscript is completely missing any discussion of a recent line of works on constrained TRPO (C-TRPO) and its proximal version (C3PO). Here, the trust regions consisting of only safe policies are designed through a mixture of the common KL geometry and a cost-dependent term (Milosevic et al. 2025, Milosevic et al. 2025).

+ Embedding Safety into RL: A New Take on Trust Region Methods, Milosevic et al., 2025
+ Central Path Proximal Policy Optimization, Milosevic et al., 2025

**Questions:**

+ Can you elaborate on the connection of SB-TRPO and C-TRPO? Both theoretically as well as empirically. I know this is a big task to ask in a short rebuttal, but as the two approaches seem very close in nature (not necessarily identical), I feel that a comparison would be very interesting and is required for publication. I am willing to raise my score, if you can incorporate the following:
	+ Comparison of obtained update directions: How does a mixture of geometries relate to a mixture of the two updates? The resulting updates can be plotted for some problems.
	+ Comparison of theoretical guarantees
	+ Most importantly: empirical comparison
+ What is the intuition behind forcing a decrease of the cost during optimization? From my understanding $L_{c, \pi_{old}} (π) \le L_{c,\pi_{old}} (π_{old}) − \epsilon$ is stronger than usual. In particular, if $\pi_{old}$ is safe, then the cost can no longer be decreased. I wonder, whether it would be more flexible to define the CMDP as usually that the cost is required to be below a certain threshold $b$ and to only require the reduction of the cost as long as it is above the threshold $b$. Note that one obtains the current formulation by setting $b=0$ and considering a non-negative cost.
+ How do you solve the trust-region formulations?

---

> ### Author Response · Authors · 2025-11-23
>
> We thank the reviewer for their valuable comments. We refer them to the general response for a more detailed justification of our focus on the zero-cost CMDP formulation. Below, we elaborate on the connection between our method and C-TRPO, clarify the rationale behind our required cost decrease, and explain how the corresponding trust-region updates are computed.

---

> > ### Author Response · Authors · 2025-11-23
> >
> > # Connection to C-TRPO
> > ## Update Directions and Geometries.
> > C-TRPO modifies the TRPO trust-region *geometry* by adding a barrier term that diverges near the feasible boundary. This yields a deformed divergence that increasingly penalises steps pointing towards constraint violation. When the constraint is violated, C-TRPO enters a dedicated *recovery phase* in which the update direction becomes the pure cost gradient; reward improvement is not pursued during this phase.
> >
> > SB-TRPO keeps the standard TRPO geometry fixed and instead modifies the *update direction* itself: the step is the solution of a two-objective trust-region problem with a required fraction $\beta$ of the optimal local cost decrease. This yields a *dynamic convex combination* of the optimal reward and cost directions. Unlike C-TRPO:
> > * there is **no separate recovery phase**,
> >  * the update direction is always a mixture of $(\Delta_r,\Delta_c)$ rather than switching geometries, and
> >  * *reward improvement remains possible in every step* whenever the reward and cost gradients are sufficiently aligned.
> >
> > This clarifies the relation between the two approaches: C-TRPO mixes *geometries* (via a KL barrier that reshapes the trust region), while SB-TRPO dynamically mixes *update directions*. In particular, during recovery C-TRPO uses $\Delta_c$, whereas SB-TRPO continues to use $\Delta$, the solution of the trust-region problem considering both reward and cost; see Fig. 1.
> >
> > Finally, the safety-bias parameter $\beta$ in SB-TRPO plays a role fundamentally different from the barrier coefficient in C-TRPO: $\beta$ determines the minimum fraction of the optimal local cost decrease required from each step, while the barrier coefficient in C-TRPO controls the strength of the geometric repulsion from the boundary.
> >
> >
> > ## Angles between Update Direction and Cost/Reward Gradients.
> > We plot the angles between policy updates and cost/reward gradients in Figure 7 in the updated paper.
> > We observe that for CPO and C-TRPO most angles between update direction and reward gradient are $90^\circ$ or larger, whilst for SB-TRPO most angles are below $90^\circ$ with a concentration around $60^\circ-70^\circ$. In the light of Theorem 2, this almost always ensures a reward improvement (for sufficiently small step size).
> >
> > On the other hand, angles between update direction of SB-TRPO and cost gradient tend to be larger than for CPO and C-TRPO, illustrating its less greedy feasibility attainment.
> >
> > ## Connection to CPO.
> > Both algorithms can be seen to generalise CPO in different ways:
> > * **C-TRPO:** show in Prop. 4.2 that the approximate C-TRPO update approaches the CPO update as the coefficient of the barrier term vanishes, recovering CPO in the limit. Besides, both CPO and C-TRPO have an explicit recovery phase when the update is infeasible.
> >  * **SB-TRPO:** As argued in Section 4.1, setting $\beta=1$ in SB-TRPO recovers CPO exactly and elegantly captures both the recovery and feasible update phases.
> >
> > ## Theoretical Guarantees.
> > Both algorithms have strong theoretical guarantees:
> > * **C-TRPO** provides
> >   - CPO-style "almost" reward-improvement guarantees for steps within the feasible region under the barrier geometry,
> >   - bounds on constraint violations for feasible steps, and
> >   - a time-continuous global convergence result under fairly strong assumptions (finite state/action spaces and "regular" policies.
> > Outside the feasible region (during the recovery phase), *reward improvement is not guaranteed*.
> > * **SB-TRPO** guarantees for sufficiently small step sizes in the quadratic-approximation regime (Theorem 2):
> >   - cost strictly decreases at every iteration (unless cost gradient vanishes),
> >   - reward strictly improves whenever reward and cost gradients are sufficiently aligned.
> > In particular, there is no recovery phase: *even if cost is high, updates still improve reward* (as long gradients are not misaligned).
> >
> > Besides, in the idealised setting of (Update 1), if neither cost nor reward improves further, the method arrives at a *trust-region local optimum for cost* and a *trust-region local optimum for the constrained problem* with the modified cost threshold (Theorem 1).
> >
> > In short, C-TRPO guarantees approximate reward improvement only within feasible steps, whereas SB-TRPO can guarantee reward improvement even for currently infeasible policies.
> >
> > ## Empirical Performance.
> > We added C-TRPO (as well as C3PO and others) to our baselines and present results in Tables 1 and 2. We can observe that C-TRPO often achieves slightly lower cost than SB-TRPO, which however typically comes at the expense of negative or significantly lower rewards, resulting in poor task performance.

---

> > > ### Author Response · Authors · 2025-11-23
> > >
> > > # Intuition behind enforcing cost decrease
> > > We refer the reviewer to the re-written derivation of our method in Sec. 4.1 for full details.
> > >
> > > We choose
> > > $\epsilon :=  -\beta \cdot \langle g_c, \Delta_c \rangle,$
> > > where$\Delta_c$ is the optimal solution to the cost-only linearised trust-region problem (Eq. (2)). This choice guarantees that the update is always feasible. For $\beta=1$, the update exactly recovers the CPO update for a zero-cost limit.
> > >
> > > The rationale behind (Update 1)-(Update 3) is that in hard-constrained problems, achieving both strict safety and reasonable task performance is challenging: existing methods typically either fail to eliminate safety violations or collapse to purely safety-driven behaviour. The update is therefore designed to ensure progress on *both* axes: it seeks reward improvement while simultaneously enforcing a guaranteed safety improvement of at least $\epsilon$.
> > >
> > > By enforcing a cost decrease less aggressively than CPO and related methods such as C-TRPO, our approach empirically helps to avoid getting stuck in poor, trivially safe but over-conservative policies. Moreover, if the current policy already achieves zero estimated cost then $g_c = 0$. Hence, $\epsilon = 0$ and no further safety improvement is required.
> > >
> > >
> > > # Computation of Trust Region Updates
> > >
> > > In practice, we implement the trust-region updates using standard TRPO-style techniques. For both the reward- and cost-directed objectives (Eqs. (2) and (3)), we estimate the gradient of the linearised surrogate and approximate the KL constraint using the Fisher information matrix. Each KL-constrained step is solved via the conjugate gradient method, avoiding explicit inversion of the Fisher matrix.
> > >
> > > The final update direction $\Delta$ to approximate (Update 3) is given by the convex combination
> > > $
> > > \Delta := (1-\mu)\cdot \Delta_r + \mu\cdot \Delta_c,
> > > $
> > > where $\mu$ is defined by Eq. (4). Recall that this update satisfies the constraints of (Update 3) (see Lemmas 2 and 3), but it is not necessarily optimal for reward improvement. In summary, this procedure adds negligible computational overhead compared to standard TRPO.

---

> > > > ### Comment · Reviewer_1Wi6 · 2025-11-24
> > > > **Thanks**
> > > >
> > > > I very much appreciate the significant updates and extensive responses and have adjusted my score accordingly. However, regarding the discussion of the relation of SB-TRPO with CPO, C-TRPO etc., I still have the following remark:
> > > >
> > > > To me, they follow a very distinct philosophy and address different problems at heart: SB-TRPO concerns a *hard constraint* problem and carries out optimization outside of this feasible set. In constrast, C-TRPO (like CPO) treats *inequality constraint* problems and aims to optimize while staying safe, hence they also report a cost-regret rather than only final cost. As such, I see them as **orthogonal approaches**, one operating in the set of safe policies, one outside. For example, they can easily be combined by replacing the CPO or C-TRPO recovery phase with SB-TRPO. Whereas this difference is to some extend discussed, this is presented as a shortcoming of CPO and C-TRPO, rather as being attributed to the fact that they are trying to achieve something different. In my view, it makes much more sense and would reflect their relation much better if they were compared as complementary and orthogonal rather as direct competitors.
> > > >
> > > > Best wishes

---

> > > > > ### Author Response · Authors · 2025-11-25
> > > > >
> > > > > We thank the reviewer for this thoughtful clarification. We agree that SB-TRPO and CPO/C-TRPO follow distinct philosophies: CPO and C-TRPO aim to optimise within the feasible set for inequality-constrained CMDPs, whereas SB-TRPO is designed specifically for hard constraints, allowing effective optimisation outside the feasible region while guaranteeing progress towards both safety and task performance. In this sense, the approaches are orthogonal and potentially complementary—we agree with the reviewer that SB-TRPO could, for example, replace the recovery phase in CPO/C-TRPO.
> > > > >
> > > > > We believe the hard-constraint setting is particularly appropriate for safety-critical applications, where any safety violation is unacceptable. In the absence of dedicated methods for hard constraints, practitioners are left with either zero cost thresholds or arbitrary positive thresholds that inadequately capture the strong safety requirements of their application. As revealed by our empirical evaluation, existing baselines often fail in this setting, either becoming overly conservative or not reliably enforcing safety, partly because they were not designed for strict zero-violation constraints. *We strongly believe that hard constraints in RL warrant wider attention by the community.*
> > > > >
> > > > > We also maintain that the experimental evaluation is informative and clearly demonstrates the benefits of our method in the hard-constrained setting on standard Safety Gymnasium benchmarks. Our goal is not to suggest that CPO/C-TRPO are inferior—SB-TRPO is not directly applicable to positive cost thresholds—but to show that SB-TRPO performs exceptionally well under zero-cost constraints.
> > > > >
> > > > > In the revised manuscript, we will clarify that:
> > > > > - SB-TRPO and CPO/C-TRPO target different notions of safety;
> > > > > - They are complementary rather than direct competitors;
> > > > > - Empirical comparison illustrates how their differing philosophies manifest in practice.
> > > > >
> > > > > We thank the reviewer again for highlighting this perspective, which helps us frame our contribution more clearly.

---

### Official Review · Reviewer_AFnS · 2025-10-31

**Soundness:** 3
**Presentation:** 3
**Contribution:** 2
**Rating:** 2
**Confidence:** 3

**Summary:**

The paper introduces SB-TRPO, a modification of Trust Region Policy Optimization (TRPO) for constrained reinforcement learning. The authors extend the TRPO objective by adding a penalized term to jointly account for reward maximization and cost minimization within a constrained Markov decision process (CMDP) framework. The goal is to improve stability and constraint satisfaction compared to existing approaches. The paper provides theoretical analysis and reports experiments on benchmark environments to demonstrate improved trade-offs between performance and safety metrics.

**Strengths:**

- Clarity and structure: The paper is well organized and clearly written, making it easy to follow the main ideas.
- Principled approach: The modification to the TRPO objective is logically motivated and mathematically consistent with the trust-region framework.
- Relevance: Addressing stability and safety in policy optimization is an important and timely problem in reinforcement learning.
- Theory and implementation: The theoretical discussion of the proposed surrogate objective adds rigor and helps frame the contribution.

**Weaknesses:**

1. Incremental contribution.
The method extends an existing algorithm (TRPO) in a relatively straightforward way by adding a penalty term to handle constraints. The conceptual novelty and theoretical differences from prior constrained TRPO or penalty-based methods are limited and should be clarified.
2. Comparative evaluation.
The experimental section would benefit from stronger and more diverse baselines. The paper appears to compare primarily with older constrained RL variants (e.g., TRPO-Lagrangian). Including a wider range of recent or more competitive baselines would make the results more convincing. See for example the methods discussed in Milosevic et al. 2024 (https://arxiv.org/abs/2411.02957)
3. Empirical depth.
The evaluation seems limited to a few standard environments. Additional experiments or analyses (such as sensitivity to penalty parameters, ablation of components, or convergence stability) would strengthen the claims.
4. Conceptual clarity.
The motivation section criticizes CMDP formulations for tolerating constraint violations but then employs the same CMDP structure. The authors should explain why this formulation remains appropriate in their setting and how their modification mitigates the stated limitations.
5. Potential bias in the surrogate.
Since the objective introduces a new penalty term, it would be helpful to discuss whether this affects convergence to the true constrained optimum or introduces bias.
6. Figures and intuition.
Some theoretical sections could be complemented by short intuitive explanations or clearer figures illustrating how the penalty affects the policy update.

**Questions:**

- Contradictory motivation: the authors criticize CMDP formulations for tolerating nonzero costs but still adopt one for their solution. This should be addressed more clearly. Can you clarify how the proposed approach addresses the CMDP limitations mentioned in the introduction?
- Does the modified surrogate guarantee convergence to the true constrained optimum, or to an approximate penalized solution?
- How sensitive is the algorithm’s performance to the penalty coefficient and trust-region size?
- How does the method behave under stricter or multiple constraints?

---

> ### Author Response · Authors · 2025-11-23
>
> We thank the reviewer for their insightful comments. In the general response, we further justify the zero-cost CMDP formulation we study, clarify the conceptual novelty of our approach, and summarise the extended empirical evaluation—including additional recent baselines such as C-TRPO and C3PO.
>
>
> Below, we address the reviewer’s specific concerns regarding potential bias in the surrogate objective, the handling of stricter or multiple constraints and the depth of ablation studies.
>
>
> # Convergence Guarantees
> We refer the reviewer to the re-written derivation of our method in Section 4.1, in particular Theorem 1, which summarises the guarantees of the general (Update 1) with the specific choice of $\epsilon$. Intuitively, cost decreases whenever possible; when no cost reduction occurs, reward is guaranteed to improve; and if neither reward nor cost improves, a trust-region local optimum for both cost and the constrained Problem 2 with a modified cost threshold is attained. Thus, the modified surrogate does not introduce any "bias".
>
>
> # "Stricter" and Multiple Constraints
> Since our formulation already enforces the strictest possible requirement, $J_c(\pi)=0$, there is no meaningful way to make the constraint ``stricter''. For multiple cost signals $c_1, \dots, c_n$, one can naturally require $J_{c_i}(\pi)=0$ for all $i$, i.e., that under any cost signal, almost surely no unsafe states are ever visited. In principle, this can be reduced to a single compound cost $c = \sum_i c_i$ with the constraint $J_c(\pi)=0$, which is theoretically equivalent. In Safety Gymnasium, for instance, different types of hazards are already encoded by default in a single compound cost signal (see Section 4.1.1).
> We note that in problems where the individual $c_i$ have very different scales additional techniques (e.g. normalisation or using indicator functions for positive costs) may be needed to balance the contributions of each cost signal, and we leave such explorations for future work.
>
>
> # Ablation Studies and Sensitivity to Hyper-Parameters
> We direct the reviewer to Sections 5.3 and C.3 for ablations of the main components and design decisions: notably the effect of the safety bias $\beta$ and eschewing critics. We have added Tables 6 and 7 using $\beta=0.7$ and $\beta=0.75$ across all tasks. The overall performance profile remains stable, demonstrating robustness to the choice of $\beta$.
>
>
> The trust-region size $\delta$ plays the same role it does in CPO and TRPO. Following standard practice, we adopt the canonical $\delta = 0.01$. Prior work has shown that this choice provides a good balance between update stability and optimization progress, and our own preliminary runs did not show clear performance variation within the conventional range ($0.005$-$0.02$).

---

> ### Comment · Reviewer_AFnS · 2025-11-27
>
> Thanks to the authors for their extensive responses and explanations. To me that makes the contribution and technical details much clearer. I would therefore increase my rating to 4. At the moment I would not go higher than that as I feel the novelty is still limited.

---

### Author Response · Authors · 2025-11-23

We thank the reviewers for their insights and comments. With their help we have revised the paper. We summarise the main changes (highlighted in olive in the revision):

* We have re-written the derivation of our method in Section 4.1, framing it as a generalisation of CPO that overcomes its over-conservatism. In particular, we have clarified the selection of the required cost decrease and included Theorem 1, which shows that the choice of safety bias $\beta$ does not affect convergence to local optima within the trust region. *We stress that the method itself remains completely unmodified.*
 * We have included a wide range of additional, more recent baselines, including C-TRPO and C3PO, which yield results consistent with those in the original evaluation.

We proceed by further motivating our zero-cost CMDP formulation, summarising additional empirical findings, and clarifying the conceptual novelty and practical benefits of our method.

We also address further issues raised by reviewers individually.


# Motivation of Zero-Cost CMDP Formulation
In many safety-critical domains, the appropriate requirement is that the agent *never* enters unsafe states (e.g. when they encode damage to an expensive robot or even accidents of autonomous cars).  In the discounted-cost CMDP formalism with non-negative costs, this requirement is expressed *exactly* by imposing the zero-threshold constraint $J_c(\pi)=0$. Introducing a positive threshold $b>0$ effectively permits some violations and, crucially, adds a hyperparameter that encodes an arbitrary notion of "acceptable risk".

The level of violation required for learning to progress varies greatly across tasks (e.g. around 17 on the hardest navigation environments, around 10 for PointPush, and well below 1 for velocity or circle tasks) as well as across algorithms, neural network architectures and optimisation hyperparameters. This makes positive-threshold CMDPs brittle, environment-dependent, and potentially misaligned with true safety objectives.
Conceptually, a positive cost threshold in the context of hard safety is undesirable because it conflates the *problem statement/specification* with an *algorithmic hyper-parameter*.

For these reasons, we focus on the *intrinsically meaningful* zero-threshold (hard-constraint) CMDP, which avoids the ambiguity of selecting $b$, removes the tuning burden associated with calibrating allowable violations, and directly matches the requirement of "almost surely no unsafe events".

Finally, we believe that the zero-cost problem is significantly under-explored in model-free Safe RL, despite being the more appropriate formulation for capturing critical safety violations in practice. By explicitly targeting this regime, our work aims to help shift attention towards this practically important but comparatively understudied problem class.


# Extended Empirical Evaluation
We have considerably extended our empirical evaluation by including the following additional baselines: CPPO-PID, FOCOPS, RCPO, PCPO, P3O, C-TRPO and C3PO.

The results are presented in Tables 1 and 2.

The expanded set of baselines yields results consistent with those presented in the submitted paper; no qualitatively new behaviour emerges, and the overarching conclusions remain unchanged.

Similarly to CPO, C-TRPO can achieve better feasibility on some harder tasks (e.g. Point Button), but their rewards are very low—mostly negative—yielding minimal task completion.
CPPO-PID and FOCOPS generally have larger constraint violations (except for Swimmer Velocity).

In Swimmer Velocity P3O and C3PO attain slightly better performance; however, our method still achieves an excellent 98% safety together with strong task performance, and we pareto-dominate P3O and C3PO on most other tasks.

Overall, the additional baselines paint the same picture: our approach is the only one to consistently achieve the *best balance of safety and meaningful task completion*.

---

> ### Author Response · Authors · 2025-11-23
>
> # Novelty and Comparison to Existing Methods
> We refer the reviewer to our re-written derivation of SB-TRPO as a generalisation of CPO overcoming its greedy behaviour in the cost-focused recovery phase, which is triggered very frequently in the setting of hard constraints.
> We proceed by highlighting our core contribution and conceptual novelty, and explaining why these lead to superior performance in practice.
>
> ## Core contribution and conceptual novelty.
>  Whilst prior work usually addresses general CMDPs, our method targets the setting with hard constraints specifically, allowing it to excel in it.
> Although our method builds on the trust-region framework of TRPO, the key conceptual novelty lies in introducing generalised update rule capturing controled, simultaneous progress on both safety and reward at every iteration.
> Specifially, the cost reduction per update is
> $
> \epsilon := - \beta\cdot\langle g_c, \Delta_c\rangle ,
> $
> where $\Delta_c$ is the optimal solution to the cost-only linearised trust-region problem Eq. (2).
> This design has three important consequences.
>
> * *No separate recovery phase.*
> Unlike CPO and C-TRPO, our algorithm never switches into a feasibility-restoration mode.
> Safety and reward are addressed within the *same* update, producing smoother learning dynamics and avoiding the abrupt, purely cost-driven behaviour characteristic of recovery phases.
> * *CPO behaviour is recovered for $\beta = 1$.*
> For $\beta=1$, the safety requirement matches exactly the zero-threshold constraint imposed by CPO, so our update includes CPO-like behaviour as a special case.
> For $\beta < 1$, the method is deliberately less aggressive in driving the policy toward feasibility, which—as our experiments show—prevents collapse into overly conservative, trivially safe states.
> * *Guaranteed improvement of both cost and reward.*
> As formalised in Theorem 2, the update guarantees
> (i) strict cost improvement whenever $g_c \neq 0$, and
> (ii) strict reward improvement whenever the reward and cost gradients are not misaligned.
> Thus, reward improvement is guaranteed even *before* feasibility is achieved—something CPO and related methods such as C-TRPO do not guarantee during its feasibility recovery/attainment phase.
>
> Moreover, compared to Lagrangian methods, which update via
> $\Delta_r + \lambda \cdot \Delta_c$ with $\lambda$ that *does not take the current reward and cost updates $\Delta_r$ and $\Delta_c$ into account* and increases monotonically under zero-cost thresholds, these methods typically either over- or under-emphasise safety.
> By contrast, we dynamically weight $\Delta_r$ and $\Delta_c$ with $\mu$ defined in Eq. (4), providing the aforementioned formal improvement guarantees for both reward and cost—*something Lagrangian methods do not offer*.
>
> In summary, our contribution is not a basic penalty modification of TRPO but a principled, guaranteed-improvement update rule for hard-constrained CMDPs that subsumes CPO as a special case while avoiding its limitations, notably its collapse into over-conservatism.
>
> ## Why CPO behaves poorly in hard-constraint settings.
> CPO attempts to maintain feasibility at every iteration. When constraints are violated, it prioritises cost reduction to restore feasibility.  In hard-constraint regimes, this often drives the policy into *trivially safe* regions (e.g., corners without hazards), which are far from the states where reward can be obtained (e.g., far from goals in navigation tasks). Escaping such regions typically requires temporary constraint violations, which CPO forbids, leading to stagnation. Moreover, CPO and related methods such as C-TRPO *do not provide reward-improvement guarantees during this feasibility-recovery phase*.
>
> ## Why our method behaves better.
> Because SB-TRPO is *less greedy about attaining feasibility*, it avoids collapsing into trivially safe but task-ineffective regions. Besides, by *not switching between separate phases for reward improvement and feasibility recovery* (as in CPO), our method exhibits markedly *smoother learning dynamics*.
>
> Moreover, Fig. 7 in the updated paper shows that SB-TRPO’s update directions align well with *both* the cost and reward gradients, whereas CPO’s updates are typically at best orthogonal to the reward gradient. By Theorem 2, such alignment *guarantees consistent progress in both safety and task performance*.
>
> In summary, SB-TRPO yields *smoother learning*, avoids collapse into conservative solutions, and enables *steady reward accumulation* over iterations. Empirically, this leads to substantially higher returns than CPO while still maintaining very low safety violations.

---

### Author Response · Authors · 2025-12-01
**Summary of Discussion Period**

Below we summarise the key clarifications arising from the discussion period that we believe further underscore the strength and significance of the paper.


1. **Practical importance and empirical strength**
Our work targets CMDPs with hard constraints, an important special case for safety-critical applications. Prior baselines address more general formulations but struggle in the strict-safety regime, either violating safety or failing to accomplish the task. In contrast, SB-TRPO demonstrates consistently strong empirical performance across *Safety Gymnasium* environments, reliably enforcing hard constraints while maintaining learning progress. *This addresses a severe and practically important limitation of existing approaches.*

2. **Contribution beyond existing methods**
SB-TRPO builds on TRPO with a generalised update that enforces a $\beta \in (0,1]$ fraction of the optimal cost decrease while simultaneously improving reward when gradients align (which empirically almost always holds). Unlike CPO/C-TRPO, there is no separate recovery phase, avoiding over-conservatism and producing smoother learning. For $\beta=1$, CPO behaviour is recovered. *SB-TRPO is therefore a principled method for hard-constrained CMDPs with improvement guarantees for both cost and reward.*

3. **Additional experiments**
We have included a wider range of baselines, video rollouts, and qualitative behaviour analysis. The new results further reinforce the central empirical message of the paper.

**Overall impact**
The discussion and revised manuscript have convincingly clarified our contributions (as acknowledged by the reviewers). SB-TRPO delivers both a genuine conceptual advance over prior work and consistently superior empirical performance in the hard-constraint setting, addressing a critical limitation of existing methods.

---

### Meta-Review · Area_Chair_pwEZ · 2025-12-30

**Summary:**

The reviewers found the paper to be clearly written, technically sound, and well motivated. At the same time, three main concerns informed the suggested decision: (1) the contribution was viewed as largely incremental, building closely on prior TRPO/CPO-style methods with limited novelty; (2) the empirical evaluation, while careful, was restricted to a single benchmark suite, leaving questions about broader applicability; and (3) reviewers expressed concerns about the sensitivity of the method to the safety-bias parameter β and its impact on learning dynamics.

**Reviewer Concerns:**

The rebuttal and additional experiments largely addressed concerns about the sensitivity to the safety-bias parameter β, and these analyses also partially strengthened the empirical evidence, mitigating some concerns about the limited experimental evaluation. However, the main concern regarding limited novelty(the first item) remains outstanding, as the method is still viewed as a principled but incremental extension of existing trust-region–based constrained RL approaches.

**Reviewer Scores:**

Based on the discussion, two reviewers indicated that they would increase their scores, likely to 4, after the authors’ clarifications and additional experiments. One reviewer stated they would keep his/her original score unchanged. Overall, even with these modest increases, the resulting scores still suggest that the paper falls short of a clear, strong endorsement.

---

### Decision · Program_Chairs · 2026-01-26

Reject